# Loss of embryonic neural crest derived cardiomyocytes causes adult onset hypertrophic cardiomyopathy in zebrafish

Sarah Abdul-Wajid[1], Bradley L. Demarest [1] & H. Joseph Yost [1]

Neural crest cells migrate to the embryonic heart and transform into a small number of cardiomyocytes, but their functions in the developing and adult heart are unknown. Here, we show that neural crest derived cardiomyocytes (NC-Cms) in the zebrafish ventricle express Notch ligand *jag2b*, are adjacent to Notch responding cells, and persist throughout life. Genetic ablation of NC-Cms during embryogenesis results in diminished *jag2b*, altered Notch signaling and aberrant trabeculation patterns, but is not detrimental to early heart function or survival to adulthood. However, embryonic NC-Cm ablation results in adult fish that show severe hypertrophic cardiomyopathy (HCM), altered cardiomyocyte size, diminished adult heart capacity and heart failure in cardiac stress tests. Adult *jag2b* mutants have similar cardiomyopathy. Thus, we identify a cardiomyocyte population and genetic pathway that are required to prevent adult onset HCM and provide a zebrafish model of adult-onset HCM and heart failure.

[1] University of Utah, Molecular Medicine Program, Eccles Institute of Human Genetics, 15 North 2030 East, Salt Lake City, UT 84112, USA. Correspondence and requests for materials should be addressed to H.J.Y. (email: jyost@genetics.utah.edu)

Neural crest (NC) cells are a prototypical stem cell population, migrating from the developing neural tube and capable of transforming into a wide range of cell types during embryogenesis[1], including cardiac outflow track in chick and mice and cardiomyocytes in zebrafish[2,3]. NC has been implicated in zebrafish, chick, mouse, and human cardiac development[4], but it is unknown whether neural crest-derived cardiomyocytes (NC-Cms) play a significant role in heart development and adult disease. The challenge has been to distinguish between primary contributions of NC to endocardial or myocardial cardiac development and sequelae caused by defects in other tissues that subsequently impact cardiac morphogenesis and cardiac function. Distinguishing between global NC versus cardiac NC phenotypes could better inform our understanding of the genetic and developmental etiology of both congenital heart disease (CHD) and adult heart disease. Previous studies have disrupted the cardiac NC population as a whole or different CHD gene candidates within the NC population and then characterized resulting cardiac phenotypes, often in the context of pleiotropic embryonic defects[5–7].

As an alternative approach to decipher NC-dependent cardiac phenotypes, we ask whether a specific population of specialized cardiomyocytes, the NC-Cms, influences cardiac development and disease, by lineage mapping and genetically ablating NC-Cms during embryogenesis. This led us to discover the roles of NC-Cms in regulating the patterning of the Notch pathway activation in cardiomyocytes during trabeculation, and in preventing predisposition to adult-onset hypertrophic cardiomyopathy.

## Results

**Genetic identification of neural crest-derived cardiomyocytes.** We and others have used several methods in zebrafish to label NC before or during migration from the neural tube region and found that a subset of labeled NC cells integrate into the heart and are co-labeled with heart-specific markers, implicating them as cardiomyocytes[2,5,8]. To address whether these cells are *bona fide* cardiomyocytes, we developed a dual transgenic that genetically marks individual cells only if they express both neural crest-specific genes and cardiomyocyte-specific genes. We call these cells NC-Cms. This dual-component system both permanently marks the cell lineage and makes it available for temporally-regulated lineage-specific cell ablation (Fig. 1a). We generated transgenic lines with a cardiomyocyte-specific driver (*myl7*) of a transcript encoding floxed GFP-Stop followed by tagRFP-fused cleavable (P2A peptide) and nitroreductase. Thus, when this transgene is recombined by Cre, it will express red fluorescence, and will allow the expressing cells to be ablated by metronidazole treatment at specific stages of development. This transgenic line was named *Cm:KillSwitch*.

In the absence of Cre-dependent recombination, *Cm:KillSwitch* expresses GFP exclusively in cardiomyocyte lineages (Supplementary Figure 1). The second transgenic component, called *Tg(sox10:Cre;cryaa:dsRed)*, is a *sox10* driver of Cre expression exclusively in the NC lineages (Supplementary Figure 2), on a vector marked for selection with cryaa:dsRed for eye expression. We crossed heterozygous *Cm:KillSwitch* adults to heterozygous adults of *Tg(sox10:Cre;cryaa:dsRed)*, selected the offspring that were double-positive for dsRed eyes and GFP hearts (+ RE + GFP, Fig. 1a). In embryos that carry both transgenes, Cre recombination removes the GFP-Stop and allows expression of tagRFP-P2A-nitroreductase exclusively in NC-Cms, not in other NC lineages and not in other non-neural crest-derived cardiomyocytes (Supplementary Fig. 3). Neighboring cardiomyocytes derived from primary and second heart fields remain GFP-positive. We found tagRFP + cells in the heart at 24 hpf (hours

post fertilization, Fig. 1b). These labeled cells increased in number until 2 dpf (days post fertilization), after which there was no significant increase in the number of NC-Cms ($27 \pm 3$ to $25 \pm 2$ cells from 2 to 4 dpf, $n = 3$ per time point, Fig. 1b–i, Supplementary movie 1). These data indicate that NC-Cms contribute to ~12% of the total number of cardiomyocytes in the embryonic 2 dpf zebrafish heart[9]. This contribution is early and achieves a steady state by 2 days of embryonic development.

The NC-Cms showed a consistent spatial distribution within the developing heart. NC-Cms localized to the apex of the ventricle and ventricle trabeculae, the outer curvature of the atrioventricular canal, and within the border of the inflow tract (Fig. 1j–m, Supplementary Fig. 3). By 5 dpf, 69% of ventricle NC-Cms were found in the trabeculae (Supplementary Fig. 3). In contrast, atrial NC-Cm spatial distribution did not appear as stereotypical as the ventricle. NC-Cm contribution to the ventricle led us to ask whether the NC-Cms are also integrated into the secondary heart field[10]. The secondary heart field marker Isl1/2 co-localized with NC-Cms in the proximal ventricle area indicating that NC-Cms also integrate to a subset of the secondary heart field (Supplementary Fig. 4).

**Ablation of NC-Cms.** To test the requirement of NC-Cms during heart development, we ablated them during their earliest appearance. Offspring from crosses of *Cm:KillSwitch* and *Tg(Sox10:Cre;cryaa:dsRed)* heterozygous parents were treated with either DMSO (0.5%, control) or 5 mM Metronidazole (MTZ) from 30 hpf to 48 hpf. Only those embryos that were double-transgenic, as indicated by dsRed-positive eyes and GFP-positive hearts (+ RE + GFP) were competent to respond to MTZ treatment and ablate the NC-Cms expressing Nitroreductase (Fig. 2a). Two controls were included: sibling embryos that were dsRed-eye negative but GFP-positive, treated with MTZ, and double-transgenic siblings (+ RE + GFP) treated with DMSO. NC-Cm-specific cell death was confirmed in + RE + GFP embryos treated with MTZ by immunostaining for activated Caspase-3, a marker of cell death. No significant cell death was observed in the two control groups (Supplementary Fig. 5).

After rinsing at 48 hpf, embryos were grown to 5 dpf and hearts were analyzed by confocal microscopy. Importantly, almost complete loss of tagRFP + cardiomyocytes was observed 3 days after the ablation period (at 5 dpf), in which an average of 13 tagRFP + ventricle cardiomyocytes in control hearts was decreased to an average of <0.5 in NC-Cm-ablated hearts (Fig. 2b–d, quantified in Fig. 2e). Occasional extruding remnants of tagRFP + NC-Cms were detected in NC-Cm ablated hearts (Fig. 2d, quantified in Fig. 2e). These results indicate efficient ablation of the initial NC-Cm population by 48 hpf, and, importantly, that no new *sox10*-expressing cardiomyocytes were produced, either by subsequent waves of NC migration or by de novo expression of *sox10* in other cardiomyocyte lineages. While a previous study implicated two waves of cardiac NC migration to the heart, an early wave and a late (post 3 dpf) wave[5], our data suggest that any late waves of NC migration into the heart do not contribute to cardiomyocytes and/or do not express the NC marker *sox10*. Persistent (past 3dpf) *sox10* expression has only been reported in peripheral glia cells in zebrafish[11], thus it is more likely that any later wave of *sox10* positive, NC-derived cells do not transform into *myl7*-expressing cardiomyocytes.

**NC-Cm-ablated hearts have aberrant trabeculation.** Given that ~12% of the total number of early cardiomyocytes are NC-Cms, it was surprising that no gross morphological phenotypes were observed in the NC-Cm ablated embryonic hearts (+ RE + GFP, MTZ treated) compared with control-treated hearts (Fig. 2b–d).

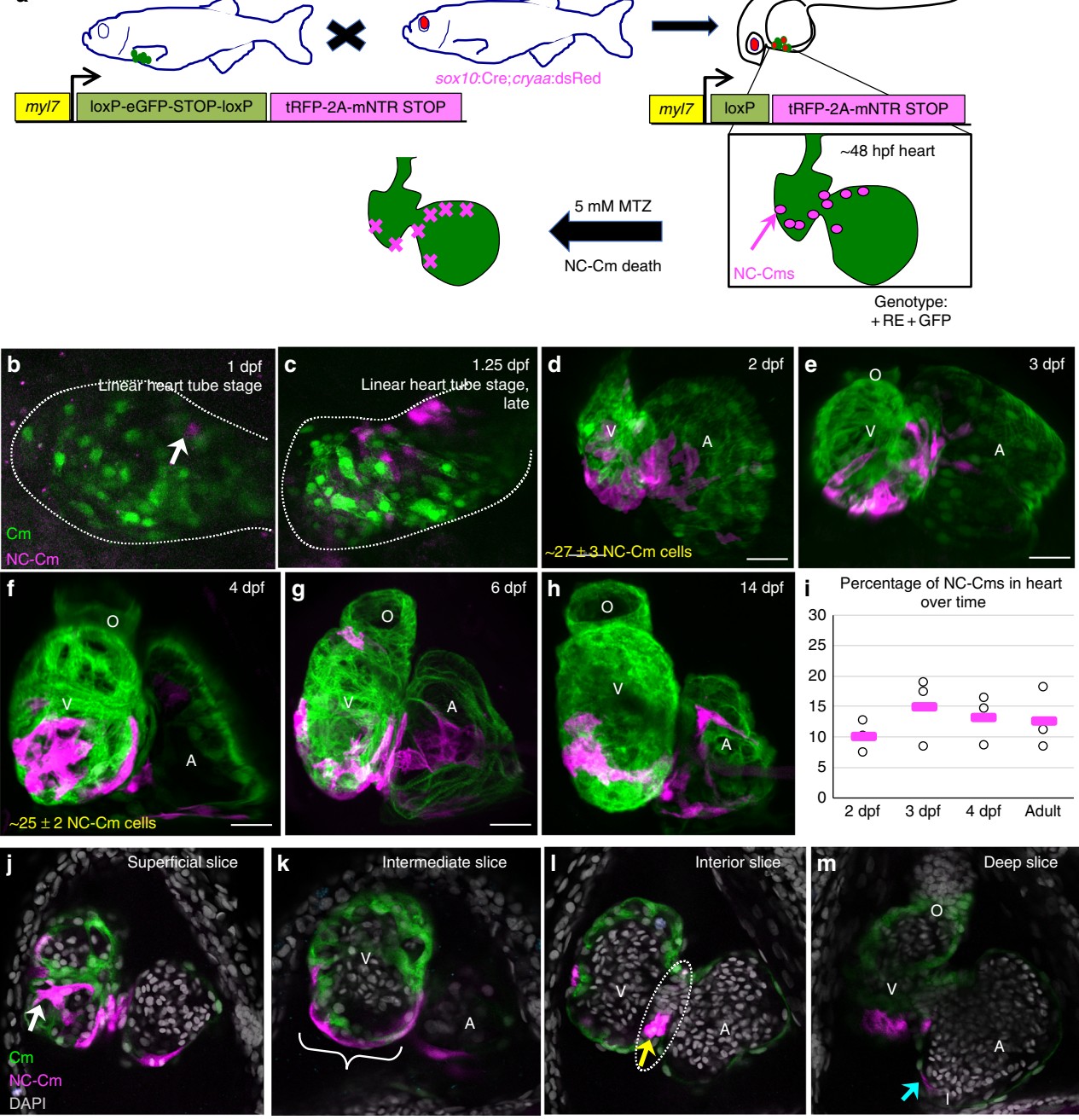

**Fig. 1** Mapping neural crest-derived cardiomyocytes (NC-Cms) during zebrafish heart development. **a** Schematic of the NC-Cm lineage labeling and ablation setup using the *Cm:KillSwitch* transgenic (*myl7*-driven transgene) crossed to the NC driver *Tg(sox10:cre;cryaa:dsRed)*. Metronidazole (MTZ) treatment causes mNTR-expressing cells to die, i.e., NC-Cms switched to express tagRFP + and mNTR. **b–h** Contribution of NC-Cms to the developing heart over time. Confocal maximum intensity images of each development stage (1–14 dpf). On average, 27 ± 3 NC-Cms were found at 2 dpf, and this number did not significantly increase by 4 dpf (25 ± 2 NC-Cms). Quantification was from confocal 3D stack images at indicated timepoints and from three individuals. Dotted line outlines heart tube. Scale bar = 25 μm. **i** Graph displays percent of total cardiomyocytes that are tRFP + (i.e., NC-Cms). Cells were dissociated from isolated hearts at each of the indicated developmental stages and analyzed by flow cytometry. After live cell gating, the sum of GFP and RFP-positive counts was deemed as the total cardiomyocyte count. RFP counts divided by total cardiomyocyte count was used to compute percentages. Circles are biological replicates at each time point and bars are the average of replicates. **j–m** Confocal slices of a 4 dpf heart from NC-Cm lineage-labeled embryos. White arrow indicates trabeculating NC-Cm. Bracket denotes the apex of the ventricle. Dashed line encircles the AV canal and the yellow arrow indicates the couple of NC-Cms found on the outer curvature of the AV canal. Blue arrow indicates the NC-Cm found at the border of the inflow tract. O = outflow tract, V = ventricle, A = atrium. Scale bar is 30 μm. Images are representative of n ≥ 3

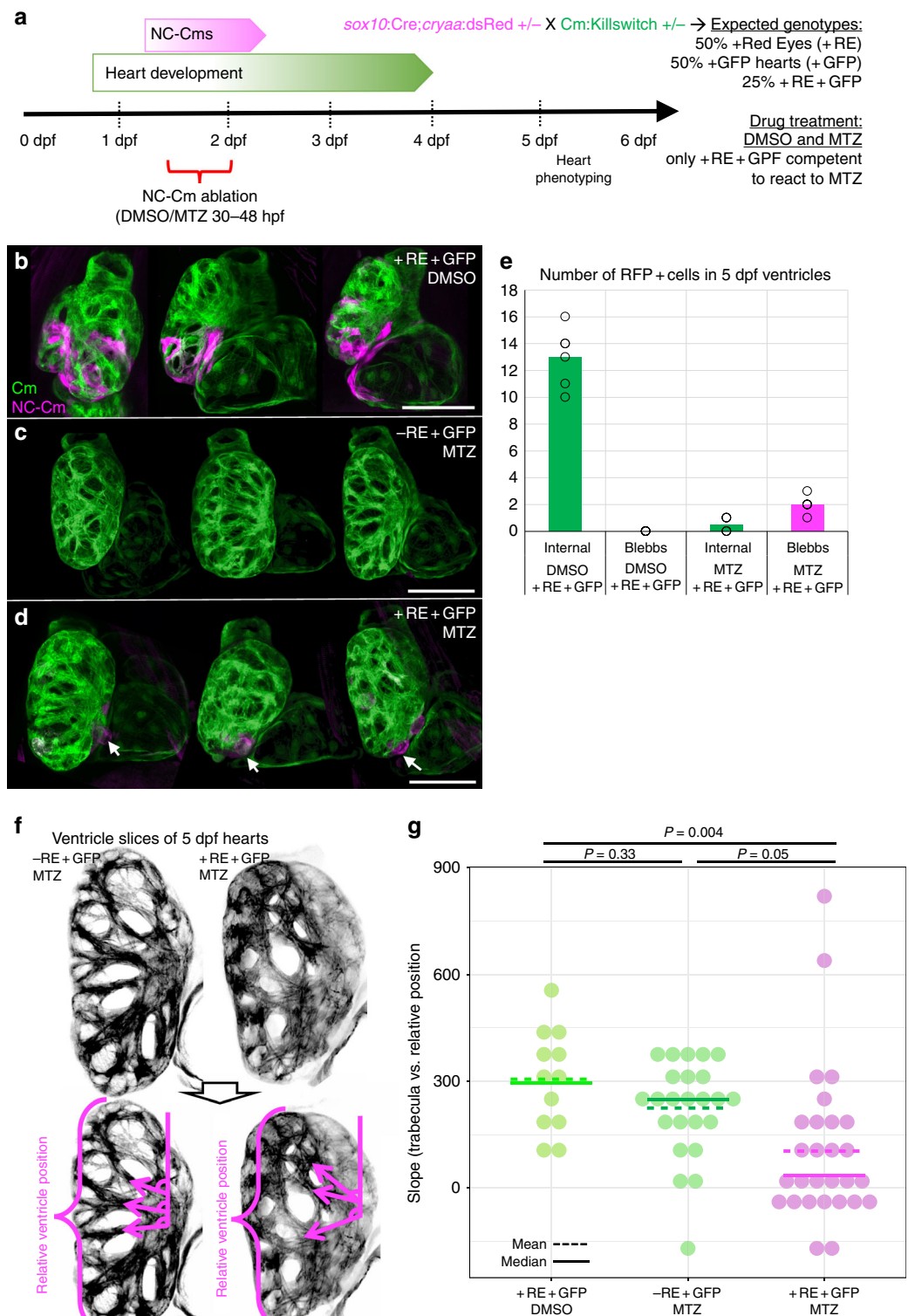

We found no significant differences in ventricle size or heart rate in NC-Cm-ablated embryos at 6dpf and juveniles at 14dpf (Supplementary Fig. 6), well beyond the age at which mutants with severe heart defects can survive[12]. However, confocal analysis at 5 dpf revealed an unusual disarray of ventricular trabeculation compared with control hearts (Fig. 2f). We quantified this phenotype by measuring the angle of the primary branch of the trabeculae as they contact the inner ventricular surface proximal to the atrioventricular canal (Fig. 2f bottom panel,

Supplementary Fig. 7). The position of each trabecula was measured relative to the anterior–posterior coordinate position (0–1.0) in the ventricle (Fig. 2f). A significant difference was found in the trabeculation pattern of the NC-Cm-ablated hearts compared with both sets of controls (Fig. 2g).

**NC-Cm relation to Notch regulation of trabeculation.** Notch signaling is an important regulator of trabeculation during heart

**Fig. 2** NC-Cm ablation alters trabeculae patterning. **a** Schematic of NC-Cm ablation protocol. *Tg(Cm:KillSwitch)* and *Tg(sox10:cre;cryaa:dsRed)* heterozygotes were crossed to generate three genotypes: *Tg(Cm:KillSwitch)* (+ GFP); *Tg(sox10:cre:cryaa:dsRed)* (+ RE); or double-transgenic (+ RE + GFP). Double-transgenic embryos were treated with DMSO (control) or MTZ from 30–48hpf to ablate NC-Cms. Sibling *Tg(Cm:KillSwitch)* (-RE + GFP) were treated with MTZ as a drug control. Embryos were phenotyped at 5 dpf. **b–d** Confocal maximum intensity projection images from three hearts at 5 dpf from each condition. NC-Cm cells (tagRFP +) were absent from the (MTZ)-treated + RE + GFP embryos compared with their DMSO treated sibling controls (**d** compared with **b**). White arrows indicate a remnant, extruding NC-Cm as a consequence of cell death. Scale bar = 100 μm. **e** Quantification of the number of tagRFP + cells in the 5 dpf ventricle ("internal") in control (DMSO + RE + GFP) and NC-CM ablated embryos (MTZ + RE + GFP). Bars are mean of individual hearts (open circles) quantified in each condition. Individual protrusions from the ventricle ("blebbs") that were tagRFP + were also quantified. **f** Trabeculation analysis of control and NC-Cm-ablated hearts at 5 dpf. Control hearts (left panels) had an array of trabeculae with primary branches arranged along anterior–posterior coordinate. In contrast, NC-Cm-ablated ventricles (from protocol in **a**) had poorly organized trabeculae (right panels). The angle of the primary branch of a trabecula and relative anterior–posterior position of the primary branch within the ventricle were measured as shown in bottom panel, magenta arrows depict primary trabecula branch; relative position in ventricle axis as represented by bracket. The position and angle of the primary trabeculae branches were measured relative to the AV canal. These data were collected for controls (-RE + GFP, MTZ treated) and NC-Cm-ablated hearts (+ RE + GFP, MTZ treated) and a slope was computed using the trabecula angle to position data for each individual heart (see Supplementary Figure 7). **g** Computed slope values for individual hearts in each treatment. Mean is indicated by the dashed line and median by the solid line. The slope measurement was significantly different for NC-Cm-ablated hearts compared with their sibling controls. *P*-values computed by TukeyHSD on ANOVA (F(62,60) = 3.31)

development[13–16]. A recent report demonstrates that Notch-activated cardiomyocytes signal to their immediate neighbors to trigger the repression of trabeculation gene-regulatory programs[13], however, the mechanisms that regulate the positioning and patterning of trabeculae are unknown. Given the aberrant organization of trabeculation patterns in NC-Cm-ablated hearts, we explored the spatial relationship of Notch signaling and NC-Cms by crossing a *sox10* reporter line *Tg(sox10:tagRFP)* with the Notch signaling reporter line *Tg(Tp1:GFP)* and examined the resultant embryonic hearts by confocal microscopy[17]. Transiently tagRFP-labeled NC-Cms were not co-incident with Notch-activated cells; NC-Cms were not found to co-express both reporters (Figs 3a–f, 3d reconstructions in Supplementary Movie 2). Instead, NC-Cms in the ventricle were immediately adjacent to Notch-activated cardiomyocytes (Fig. 3a–f, white arrows vs. arrowheads). This result suggests that the NC-Cms are the cardiomyocytes that provide Notch ligand and participate in trabeculation, as illustrated in Fig. 3m. Therefore, we asked whether Jag2B, a Notch ligand known to be expressed in cardiomyocytes and required for normal trabeculation[13,18,19] is disrupted in NC-Cm-ablated hearts.

Isolated embryonic 4 dpf hearts from NC-Cm-ablated fish and their sibling controls were used for quantitative PCR analysis of *jag2b*. Jag2B was downregulated in NC-Cm-ablated hearts, while *myl7* and *nrg2a* expression (controls) were not significantly changed compared with treated control hearts (Fig. 3g). We then investigated whether *jag2b* expression is enriched in NC-Cms during trabeculation at 3 dpf. Using fluorescent in situ hybridization, we found a significant association of *jag2b* transcript in NC-Cms compared with other cardiomyocytes (Fig. 3h–k, Supplementary Fig. 8). Quantification of these in situ results indicates jag2b transcript is enriched in NC-Cms, with lower levels of expression in other cardiomyocytes (*p* = 0.0011; Fig. 3l). Together, these data lead us to propose that NC-Cms are critical for the correct patterning of ventricular trabeculation by providing a significant and stereotypically positioned source of Jag2B to neighboring cardiomyocytes (Fig. 3m).

**NC-Cm ablation results in adult hypertrophic cardiomyopathy.** Ablation of NC-Cms resulted in a consistent defect in trabeculation patterning in embryos and juveniles but did not diminish viability, and the embryos grew to adulthood (*n* = 39/45 for MTZ-treated –RE + GFP controls compared with *n* = 44/45 for MTZ-treated + RE + GFP siblings). The effects of altered embryonic trabeculation on adult cardiac structure and physiology have not been reported. Whole mount fluorescence imaging

of adult control hearts found large patches of tagRFP + NC-Cms in the apex of the heart (Fig. 4a; Supplementary Fig. 10B, C), indicating that NC-Cm lineages persisted into adulthood, with similar topological distribution seen in embryonic hearts. Flow cytometry of dissociated adult ventricles indicated that tRFP + NC-Cms were 12.5 ± 4.9% of the total ventricular cardiomyocyte population (*n* = 4, Supplementary Fig. 9A). The hearts in which NC-Cms were embryonically ablated (Fig. 4c, + RE + GFP, MTZ treated) had negligible tagRFP fluorescence in the heart as quantified by FACS (0.8 ± 0.6%, *n* = 5, Fig. 4d and Supplementary Fig. 9B), indicating that *no subsequent* (post 2 dpf) contributions of NC cell lineages persisted in the heart into adulthood, and that de novo *sox10* expression did not occur after the initial embryonic NC-Cm contribution (Fig. 4c image is of the most extreme example of remnant tagRFP fluorescence in embryonic NC-Cm ablated adult hearts n = 20 examined). Remarkably, sections of the NC-Cm-ablated hearts (Fig. 4g; Supplementary Fig. 10D) revealed a massive increase in muscular tissue of the ventricle compared with controls (Fig. 4e–f, Supplementary Fig. 10B, C), more substantial than that predicted by the subtle alterations in the patterning of embryonic trabeculation. Ventricle muscular tissue hypertrophy was quantified by determining the percentage of area in a ventricle section covered in GFP + cardiomyocyte tissue, which was ~70% in control hearts and ~87% in NC-Cm-ablated hearts (Figs. 4h, 70.9%, 72.1%, and 87.2% mean values in DMSO + RE + GFP(CTL), MTZ -RE + GFP (CTL), and MTZ + RE + GFP (NC-Cm ablated), respectively). This measurement is inversely correlated with the luminal area that is available for blood flow through the ventricle, which was decreased ~twofold, from 30% in controls to 13% in NC-Cm-ablated hearts.

To assess whether this zebrafish hypertrophy phenotype is analogous to hypertrophic cardiomyopathy (HCM) in humans[20], we quantified cardiomyocyte number and size in NC-Cm-ablated adult hearts compared with control siblings using multiple approaches. Using Mef2 antibody staining along with DAPI to specifically demarcate cardiomyocyte nuclei in ventricle sections of the adult hearts, we found no significant difference in the number of cardiomyocytes in NC-Cm-ablated hearts compared with their control siblings (Supplementary Figure 11). To determine if cell size was altered, ventricles from NC-Cm ablated and control adults were isolated, dissociated into single cells and cultured in chamber slides for 24 h to allow adherence of dissociated single cardiomyocytes. The cells were then fixed and stained for GFP and DAPI, and GFP-positive cardiomyocytes were imaged and analyzed for cell morphology (Fig. 4i–j, Supplementary Movies 3 and 4). Cardiomyocytes from NC-Cm

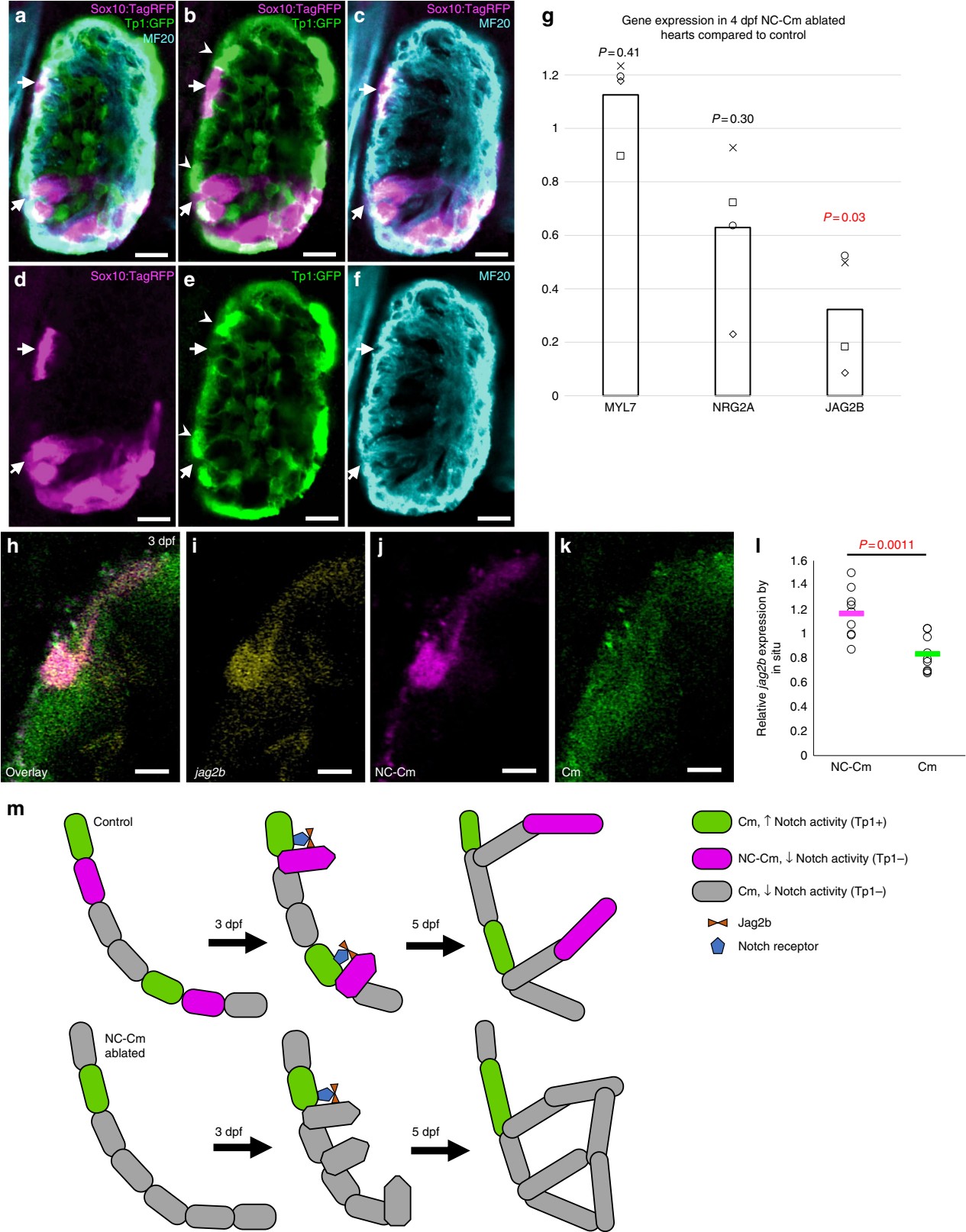

ablated ventricles were significantly larger in area and length than their sibling control cardiomyocytes (quantified in Fig. 4k–l, *n* ≥ 25 cells per condition, amalgamation of *N* = 4 repeats, n = 3–5 hearts, p = 0.02 for area and p = 0.03 for length measurements). The observation of increased cardiomyocyte size and comparable

cardiomyocyte numbers in NC-Cm-ablated hearts was confirmed by flow cytometry analysis of dissociated ventricles from NC-Cm-ablated adults and sibling controls (Supplementary Fig. 12), in which the percent of large-size-Cms increased from 56.5% in control siblings to 74.1% NC-Cm-ablated adult hearts. Together,

**Fig. 3** NC-Cm cells regulate Notch signaling during trabeculation. **a–f** Ventricle sections of a 3 dpf embryo from a *sox10:tagRFP* transgenic line crossed with the Notch reporter *Tp1:GFP* line, immunostained for the cardiomyocyte marker MF20. NC-derived lineages (tagRFP) did not show high levels of Notch response (GFP). Arrows indicate trabeculating NC-Cms that were next to a Notch-activated cardiomyocyte (arrowheads). Scale bar = 10 μm. **g** qPCR gene expression of isolated 4 dpf hearts from NC-Cm-ablated and control siblings. Values are delta delta Ct computations. Bars represent mean of four biological replicates. Points are individual experiments. Delta Ct values used to compute standard T test significance (*P*-values shown) between NC-Cm ablated and control delta Ct values. **h–k** Jag2b fluorescent in situ hybridization in a magnified ventricle section of 3 dpf NC-Cm lineage-labeled embryos. Scale bar = 5 μm. Image is representative of *N* = 17 embryos imaged and analyzed in probe positive in situs compared with probe negative embryos imaged and analyzed. **l** Quantification of *jag2b* fluorescent in situ signal in individual NC-Cm and Cm ventricle cells. Relative intensity was computed by the average intensity of all *jag2b* expression in the heart cells and then comparing with individual NC-Cm tRFP + and Cm GFP + cell intensities. Circles are individual cells from *N* = 3 embryos from three independent in situ experiments. Bar = mean. Standard T test was used for *P*-value. **m** Schematic model in which NC-Cms provide spatial patterning of trabeculation using Notch signaling components. During the transition to initiate trabeculae, NC-Cms provide a significant source of *jag2b* expression that triggers its neighbor Cm to repress protrusion and trabeculation initiation. This results in evenly, spatially distinct trabeculae branches at 5dpf. However, without NC-Cms and their major source of *jag2b* expression (although not exclusive as indicated by the remaining Cm expressing *jag2b* at 3dpf), more Cms are poised to protrude and create trabeculae, yielding conjoined branches and poor trabeculae spacing by 5 dpf

these results indicate that increased cardiomyocyte size contributes to the hypertrophic cardiomyopathy phenotype in adults raised from NC-Cm-ablated embryos.

**Mutants of jag2b have adult hypertrophic cardiomyopathy**. We hypothesized that if the Jag2B signal that is embryonically supplied in the ventricle by NC-Cms was critical for the development of the NC-Cm-adult hypertrophy phenotype, *jag2b* mutants might phenocopy NC-Cm ablation. Surprisingly, *jag2b* homozygous mutants are viable as adults, and we found a striking phenocopy of the hypertrophic ventricle of NC-Cm-ablated adults (Fig. 5). Moreover, a heterozygous *jag2b* mutants also displayed hypertrophic cardiomyopathy (Fig. 5b), although not as severe as homozygotes (quantified in Fig. 5d), suggesting that haploinsufficiency of *jag2b* is also causal for this phenotype.

**Embryonic NC-Cm ablation predisposes adults to heart failure**. Because adult hearts from embryonic NC-Cm ablation were dramatically hypertrophied, with diminished lumen volume for blood flow, we asked whether this HCM impacts adult cardiac function. NC-Cm-ablated adults and sibling controls were subjected to a cardiac stress test in a swim tunnel assay, in which adult fish are challenged with step-wise increases in water speed, analogous to step-wise increases in treadmill speed in stress-tests of human adult cardiology patients (Fig. 4m–n). The swim tunnel assay measures the critical water speed at which individual fish fatigue and stop swimming (Ucrit, Supplementary Movie 5)[21,22]. NC-Cm-ablated adults performed significantly poorer in the swim tunnel assay than their sibling controls and fatigued at much lower Ucrit values (quantified in Fig. 4o). These results indicate that ablation of NC-Cm lineages during the first days of life can predispose adults to performance-induced heart failure.

## Discussion

Overall these findings demonstrate previously unknown roles for NC-Cms, using unique lineage labeling and genetic ablation approaches. Importantly, the hypertrophic cardiomyopathy and heart failure in adults and aberrant trabeculation patterning in embryos can only be attributed to the post-migratory NC that have converted to *bona fide* cardiomyocytes. Previous attempts to analyze the consequences of NC ablations reported changes in embryonic ventricle morphology, heart rate, and other defects that we did not observe[3,5,23]. Those reported effects were likely due to secondary effects of perturbation of other NC-derived lineages that contribute to other embryonic structures such as aortic arches or endocardium, resulting in pleiotropic phenotypes that can have secondary effects on heart function[5,8,23].

The outcomes of ablating NC-Cms stand in striking contrast to previous studies that arbitrarily ablated large numbers of embryonic ventricular cardiomyocytes and reported no consequential effects on subsequent embryonic heart regeneration, function, and trabeculation[24,25]. Two days after ~12% loss of cardiomyocytes by NC-Cm ablation, we found no significant difference in cardiomyocyte quantity (Supplementary Fig. 13), consistent with cardiomyocyte cell number homeostasis in adulthood (Supplementary Fig. 11, 12). Thus, in response to NC-Cm ablation, the embryonic heart undergoes a regenerative process to provide a homeostatic number of cardiomyocytes, presumably from the non-NC-derived cardiomyocyte population. However, this regenerative process fails to protect from adult hypertrophic cardiomyopathy and heart failure, indicating that the NC-Cms are a unique cardiomyocyte population that cannot be regeneratively replaced by other cardiomyocytes. While some of the trabeculae mispatterning could be directly due to loss of NC-Cms that contribute to trabeculae, overall cardiomyocyte number is rapidly returned to normal, and the enrichment of *jag2b* in NC-Cms and activation of Notch signaling in neighboring cells suggests a cell non-autonomous pathway that contributes to trabeculae patterning. NC-Cms supply a required, innate function, most likely mediated by *jag2b*, that regulates trabeculae patterning in the embryonic ventricle and protects from adult-onset HCM and heart failure.

Our findings clarify the roles of Notch-regulated trabeculation in the ventricle during embryonic development. Previous models did not provide insight into the patterning of cardiomyocytes to express Notch signals, which then subsequently trigger neighboring cells to Notch activation and suppression of trabeculation[13]. We found that NC-Cms are stereotypically distributed in the ventricle, express *jag2b* and are adjacent to Notch-responding cardiomyocytes. Ablation of NC-Cms results in diminished *jag2b* expression and correspondingly altered patterning of trabecula. Genetic loss of *jag2b* results in adult hypertrophic ventricles that mimic embryonic NC-Cm ablation. From these findings, we propose that NC-Cms serve as a pre-specified source of topologically patterned Jag2B presentation in the 3 dpf ventricle, which then impacts the spatial patterning of trabeculation. Thus, in the absence of NC-Cms, normally patterned presentation of Jag2B is lost and trabeculae are disorganized. While NC-Cms are not known to comprise a significant portion of the mammalian cardiomyocyte population[26–28], their roles in the patterning of trabeculation and in adult heart function have not been explored.

Our study provides two models of adult HCM and heart failure in zebrafish: disruption of NC-Cms during embryogenesis and genetic loss of *jag2b*. In humans, at least 23 genes have been implicated in HCM[29]. However, our finding that zebrafish *jag2b* mutants have adult cardiomyopathy suggest a

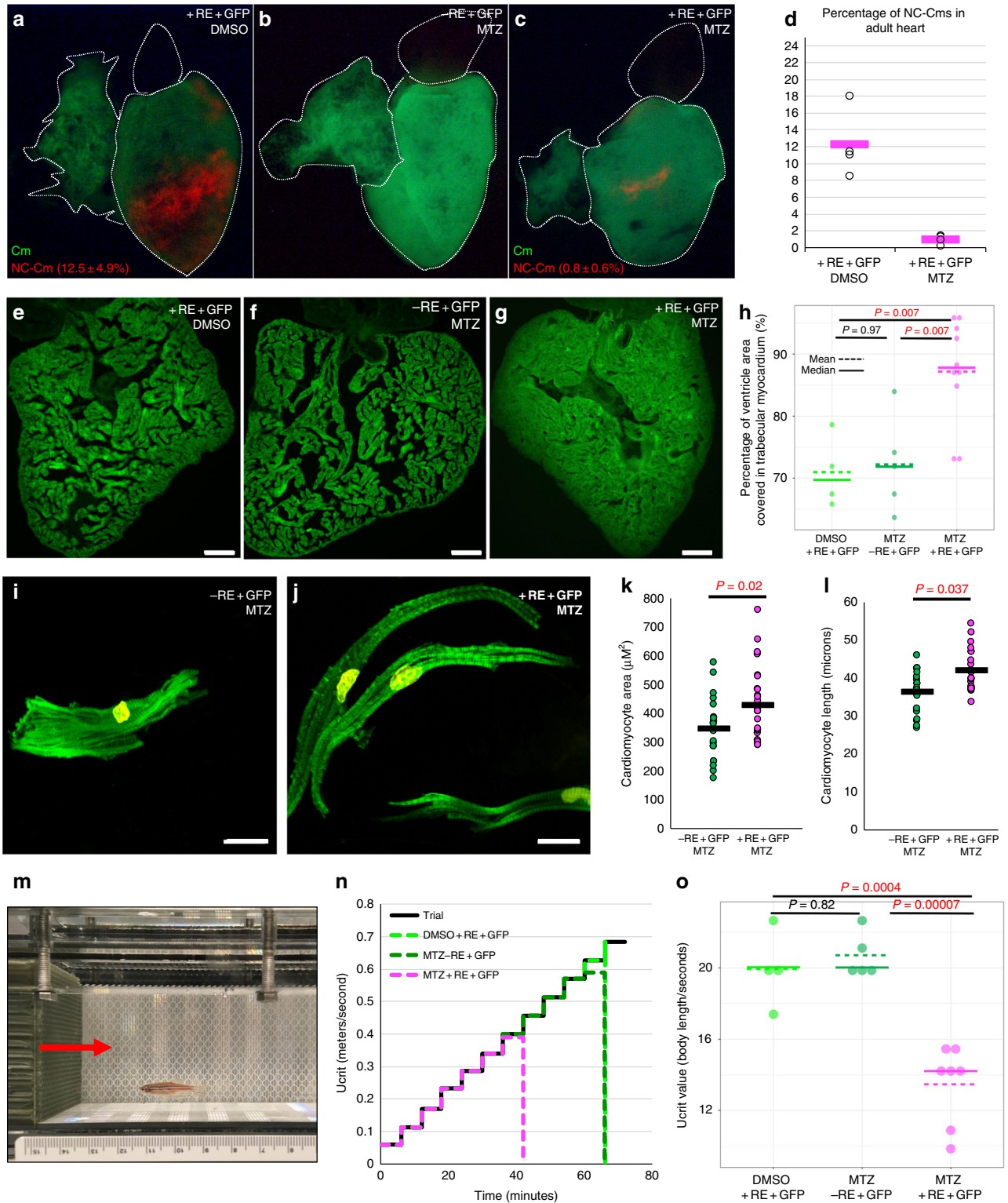

pathway that has not been implicated in human HCM. We speculate that in humans, if just NC-Cms or the genetic regulatory pathways expressed therein are specifically perturbed during embryogenesis, by mutation, developmental defects or environmental influences, individuals could have subclinical defects that only become apparent as stress-induced adult-onset heart failure. Further studies of the genetics and developmental regulatory mechanisms of NC-Cms will inform our understanding of the embryonic etiologies and genetic pathways that predispose humans to adult-onset HCM and heart failure.

## Methods

**Zebrafish husbandry**. Zebrafish were housed, and protocols were performed in compliance with ethical standards reviewed and approved by University of Utah IACUC, protocols 18–05006 and 15–06004. The AB genetic background was used for all experiments and lines generated. For embryo studies, sex was indeterminant. For adult studies, sex is indicated in figure legend.

**Fig. 4** Ablation of embryonic NC-Cms results in adult-onset hypertrophic cardiomyopathy and heart failure. **a–c** Whole-mount fluorescent images of adult hearts from embryonic NC-Cms ablation experiments (protocol in Fig. 2a). Red numbers represent percent of NC-Cms relative to total cardiomyocytes (GFP + and RFP +), not total heart cells, quantified by flow cytometry (FACS) of dissociated adult hearts (Supplementary Figure 9). **d** FACS quantification of NC-Cms. Dots are individual hearts from each condition and bars = mean of individuals in each group ($n = 4 + RE + GFP$, DMSO and $n = 5 + RE + GFP$, MTZ) **e–g** Fluorescent microscopy sections of hearts from sibling individuals as in A-C. scale bar = 100 μm. **h** Quantification of area of ventricle covered in cardiomyocytes from sections similar to **d–f**. Dots represent sibling individuals from each condition pooled from biological replicates. Solid line = median, dashed line = mean. $P$-values computed by TukeyHSD on ANOVA ($F(2,16) = 9.48$). **i-j** Examples of single-cell cardiomyocyte morphology from dissociated control (**i**) and NC-Cm-ablated adult ventricles. Cells were cultured in chambers for 24 h to allow cell attachment, then fixed and stained with anti-GFP and DAPI and visualized by microscopy. Scale bar = 10 μm. **k** Quantification of individual cardiomyocyte area from chamber cultures as in "**i, j**"; $n \geq 25$ cells per sample. P-value from standard T test. **l** Quantification of individual cardiomyocyte lengths from chamber cultures as in "**i, j**". $P$-value from standard T test. **m** Example still image from movies of swim tunnel assays of individual male zebrafish. Red arrow indicates water current direction. **n** Swim trial assay provided incremental water speed increases of 0.05 m/s every 6 min (solid black line). Average assay results for DMSO control (light green, $n = 4$), MTZ control (dark green, $n = 5$) and NC-Cm-ablated (magenta, $n = 8$) adults. Data are amalgamated from $n = 5$ biological replicates. See Supplementary Movie 2. **o** Individual Ucrit results normalized to body length from assay in "**m**". See Methods for Ucrit calculation. Dots represent individual males from each condition. Solid line = median, dashed line = mean. $P$-values computed by TukeyHSD on ANOVA ($F(2,13) = 24.65$)

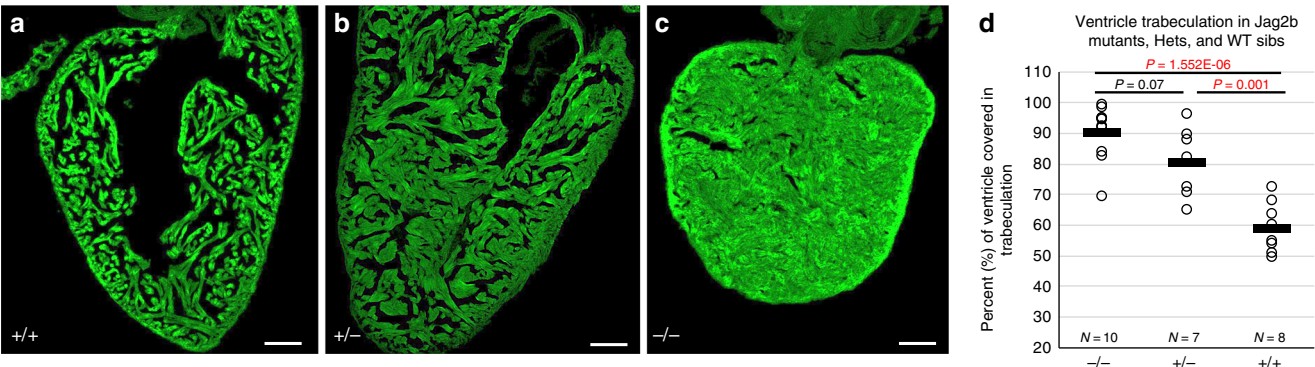

**Fig. 5** Jag2b mutant adults phenocopy hypertrophic cardiomyopathy of the NC-Cm ablated adults. **a–c** Microscopy sections of adult ventricles from *jag2b* wild-type (+/+), heterozygous (+/-), and homozygous (-/-) mutants. Sections were stained with phalloidin-488 and imaged. Scale bar = 100 μm. **d** Area of ventricle with 488 + trabecular myocardium was quantified as in 4 H and tested for significance by two-way T tests

**Transgenic line generation**. The p5E-entry clone for the 7.2 kb *sox10* promoter has been extensively characterized by Bruce Appels' lab[30–33]. The *p5E-sox10* clone was used with zebrafish codon optimized Cre, called pME-iCre (gift of K.Kwan lab), *p3E-cryaa:dsRed* and pDestpA to generate a final construct, via Gateway LR recombination technology: *sox10:iCre;cryaa:dsRed*. To generate *myl7:loxP-eGFP-loxP-tRFP-2A-mNTR* (called *Cm:KillSwitch*,), the *p5E-myl7*, *pME-floxedGFP*, *p3E-tRFP-2A-mNTR*, and *pDestpA* were recombined with Gateway LR recombination. *P3E-tRFP-2A-mNTR* was generated by PCR isolation of the 2A-mNTR sequence from the plasmid *p3E-YFP-2A-mNTR* (gift of J.Mumm lab)[34], followed by a fusion PCR with the tagRFP coding sequence. Final constructs were sequence verified and injected at 25–30 ng/μl with 30 ng of Tol2 mRNA into single-cell stage AB embryos[35]. At least three founders were screened, verified for similar expression patterns of each transgene and outcrossed to AB adults to propagate the line. *Tg (Cm:KillSwitch)zy62* and *Tg(sox10:iCre;cryaa:dsRed)zy70* heterozygotes were used in all experiments described.

**Transgenic line manipulation**. To ablate NC-Cms and demonstrate effectiveness/specificity of the *Cm:KillSwitch* line, different doses/incubation times of metronidazole (MTZ, Sigma cat. No. 46461) treatment were tested, ranging from 5 mM at 30 hpf–48 hpf to 10 mM MTZ treatment at 48 hpf–56 hpf. For all reported experiments, MTZ treatment was performed at 5 mM MTZ 30 hpf 48 hpf. MTZ stock was resuspended in DMSO at 1 M concentration followed by dilution in E3 embryo media to achieve the correct dose. MTZ stock was stored at 4 °C in the dark but not used older than a week after resuspension. The MTZ treatment regimens tested resulted in increased, specific and detectable cell death immediately after treatment (Supplementary Figure 2).

**Flow cytometry**. Adult ventricle or whole hearts were dissected out of anaesthetized fish and placed into cold HBSS + 1%FBS media. Hearts or ventricles were then allowed to pump for a few minutes in order to release any blood and squeezed gently with forceps to remove blood. For analyzing only ventricles, the atrium and outflow tract were manually dissected away with forceps. Otherwise whole hearts or ventricles were cut into pieces and placed into Liberase DH (1 mg/ml) containing HBSS + 1% FBS solution similar to the previously described protocol[36]. They were then placed in a 28 °C shaking incubator to dissociate for ~15–20 min. Pipetting every 5 min was used to aid in dissociation. ~200uL of dissociation

solution was used for 1–3 hearts. Dissociated samples were spun down at 300xg for 5 min, supernatant removed without disturbing the cell pellet and then resuspended with 350 μl of HBSS + 1% FBS, placed on ice, incubated with DAPI for a viability analysis, and processed for flow cytometry analysis on a BD FACS Canto.

**Immunofluorescence and microscopy**. For endogenous transgenic fluorescence detection and imaging, embryos were incubated in E3 media with PTU addition to prevent pigment formation. Embryos were briefly treated with 0.5 M KCl to relax hearts and then immediately fixed in 2% PFA + PBS for 1 h at room temperature. Embryos were washed and mounted in low melt agarose for microscopy. Three-dimensional images were acquired on a Zeiss LSM 880 Airyscan under fast mode and 20X magnification.

Antibody staining for anti-active-caspase3 was done according to the published protocol[37]. Briefly Rabbit-anti-active-Caspase3 antibody was used (1:200, BD Pharmingen 559565) with Chicken anti-GFP (1:1000, Aves labs) in 4% PFA-fixed embryos permeabilized with 100% cold methanol for 2 h at −20 °C. Washes used PBS + 3% Tritonx100. Antibody staining for Isl1/2 in the NC-Cm-labeled embryos was carried out as described[10]. Embryos were fixed in 2% PFA + PIPES buffer and incubated with Rabbit anti-tRFP (1:200, Life Technologies R10367) and Chicken anti-GFP. Antibody staining for MF20 (DSHB) was carried out as described[38]. Antibody staining for Mef2 (Abcam 64644) on ventricle sections was carried out on 4% PFA + PBS, fixed adult hearts that were cryosectioned into 10 μm sections. Sections were boiled in citrate buffer for ~40 min, washed with PBS + 0.3% Triton x100 (PBT), blocked with PBT + 5% goat serum, 1% DMSO, and 5 μg/ml BSA. Antibody staining was carried out at 1:200 in blocking solution overnight at 4 °C. Washes used PBT and secondary antibody staining utilized AlexaFluor 568 goat anti-rabbit (1:500, Invitrogen) in blocking solution.

Chamber cultures of dissociated adult ventricles were carried out by dissociating pools of isolated ventricles ($n = 3–5$) similar to the flow cytometry protocol above. The resuspension was then distributed into a single chamber of a Lab-Tek 8-chamber, chamber slide and incubated for 24 h at 28 °C to allow settling and adherence of cells. To fix cultured cells for microscopy, media was largely removed but never left completely dry and 4% PFA + PBS was added carefully to not disturb adherent cells and incubated for 10 min. Chambers were washed in PBS and processed for staining with GFP antibody as in above methods. Images were acquired on Zeiss LSM 880 or Leica compound fluorescent scope.

**Microscopy equipment and settings**. Zeiss LSM 880 Airyscan confocal images were acquired under "FAST" parameters with a 20X Plan-Apochromat (0.8) lens. Hearts were first zoomed-in on and brought into focus and set for a Z-stack and then super-resolution settings were utilized for all acquisitions (varied slightly based on positioning of the heart in each embryo for each experiment). Airyscan processing was then carried out on each acquisition using the Zen(Black) analysis software. Gain, laser percentage, and voltage were set independently for each experiment and kept constant through all sample acquisitions.

Acquisition information for embryonic 5 dpf heart analysis:
Three channels: (488,568,633 (MF20-staining)
Image bit depth: 16 bits/pixel
Resolution:2000–2600 × 2000–2600 (varied based on positioning of the heart)
Acquisition information for embryonic 2–5 dpf heart timecourse analysis:
Three channels: 405 (DAPI), 488,568
Image bit depth: 16 bits/pixel
Resolution: 2000–2600 × 2000–2600 (varied based on positioning of heart)
Acquisition information for embryonic 3 dpf TP1 and sox10 analysis:
Four channels: 405 (DAPI), 488,568,633 (MF20-staining)
Image bit depth: 16 bits/pixel
Resolution: 2000–2600 × 2000–2600 (varied based on positioning of heart)
Acquisition information for embryonic 3 dpf jag2b in situ analysis:
Three channels: 405 (DAPI), 488, 568, 633
Image bit depth: 16 bits/pixel
Resolution: 2000–2600 × 2000–2600 (varied based on positioning of heart)

**In situ hybridization**. Double colorimetric in situ hybridization was carried out as previously described[39]. Anti-sense Fluorescein-labeled probe for *myl7* was generated as described previously[40]. The probe for *cre* was generated by first PCR amplifying the sequence from the pME-iCre clone with M13 primers. This template was then used in a T7 RNA polymerase reaction with DIG-UTP to create the DIG labeled anti-sense *cre* probe. Double in situs were imaged on a Leica compound fluorescent scope.

Jag2b fluorescent in situ hybridization in combination with immunofluorescence for GFP and tRFP was carried out as previously described[13]. In situs were imaged on Zeiss Airyscan. To quantify cellular levels of *jag2b*, RFP and GFP areas of confocal max intensity projections were first delineated as ROIs in Image J. These ROIs were then assessed for mean *jag2b* intensity signal as well as the total ventricle area signal intensity. Each ROI was then given a relative value to the total ventricle *jag2b* signal and graphed in Fig. 3d.

**Image analysis**. Imaris software (v8.4.1) was used to reconstruct 3D microscopy images and count cell numbers. Imaris "surfaces" was used to analyze the volume of NC-Cm contribution by generating a surface for the entire RFP channel and assessing the volume compared with the volume of the GFP channel + RFP channel (combined they were considered the whole Cm volume of the heart). The "clipping plane" feature of Imaris was used to view individual trabeculae in embryonic ventricles and generate angle measurements using the "measurement points" feature. Only ventricles with greater than three measurable angles were used for analysis.

Trabeculae angles and relative heart position data were gathered for embryonic ventricles from Imaris analysis. These data were input into R programming interface to compute a slope value for each heart measured. Slopes were computed using the "lm" function (linear regression model) in R, on individual ventricle measurements. Individual heart data for angle by position measurements are displayed in Supplementary figure 7 along with their computed slope and regression value.

Measurements of cardiomyocyte coverage in adult ventricle sections were generated using Image J software. GFP fluorescence, demarcating cardiomyocytes in the myocardium, in adult ventricle sections were thresholded for a uniform value of intensity and then applied across all samples. A uniform area ROI selection was used across all section samples to generate the total area and then area of the ROI covered in GFP fluorescence was used as the percent of ventricle area covered in cardiomyocytes.

Image J was used to quantify the number of Mef2 +, DAPI + nuclei within the adult ventricle. Myocardium was outlined and selected manually to create an ROI and Mef2 +, GFP + nuclei were counted within each ROI, followed by a measurement of the ROI area. The number of nuclei were divided by the area in millimeters squared to tabulate values graphed in Supplementary Figure 7A.

For analysis of cardiomyocyte size in chamber cultures of dissociated ventricles, Image J was used to manually outline the GFP + cardiomyocyte images acquired by compound fluorescent microscopy. DAPI was used to confirm a single cardiomyocyte was being analyzed and outlined for area measurements. To calculate length and width of cardiomyocytes, the longest line was drawn from edge to edge length or width wise and then measured in micron units. Width measurements were not significant between NC-Cm-ablated cardiomyocytes and sibling controls and area and length measurements are graphed in Fig. 4n and Supplementary Figure 10.

**Swim trials**. A Loligo Systems swim tunnel setup was used to test individual adult males greater than 4 months of age for swim trial performance. Swim speeds in meters per second were based on calibrated instrument setting measurements. Incremental step sizes of 6 min were used ($t_s = 360$ s) for computation of the Ucrit value. Measurements were calculated as described previously[21,22]. Briefly, Ucrit = $U_f + U_s$ x ($t_f$ / $t_s$) where $U_f$ is the speed of the water at fatigue, $U_s$ is the water speed step (0.05 m/s), and $t_f$ is the time spent on the last step of the trial before fatigue. Fish were considered fatigued based on their inability to remove themselves from the mesh tunnel end for greater than 4 s. Body length normalization of Ucrit values was generated by measuring the length of the fish. Data shown are a compile of multiple biological replicates and sibling cohorts.

**Quantitative PCR**. In total, 4 dpf hearts were extracted from treated and control siblings from the *Cm:KillSwitch* cross to *Tg(sox10:Cre;cryaa:dsRed)* as described previously[41]. These hearts were immediately lysed in trizol and processed for RNA using a Zymo mini RNA kit. The total RNA was used for cDNA synthesis with BioRad iScript 5X master mix. Subsequent cDNA was used in multiplex qPCR reactions with the following gene primers: *rpl11, myl7, jag2B, nrg2A*. Primer and probe sequences are listed in Supplementary Table 1. Delta Ct calculations were normalized to *rpl11* expression levels and control sibling expression levels.

**Statistics**. R graphic programming was used to generate the dot plots in Figs. 2g, 4h, o. ANOVA tests were run on dot plot data to test significance.
Standard two-component (two-way) T tests were performed on all other data.

## Data availability
No large datasets were generated for this article. Data supporting the findings in this study are available within the article and Supplementary information. Transgenic zebrafish and DNA constructs described in this article, as well as additional information, are available from corresponding author upon reasonable request.

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

## Acknowledgements

We gratefully acknowledge Dr. Jerald Johnson, BYU, for lending us a swim tunnel for these studies. We thank Dr. Jeffrey Mumm and Dr. Kristen Kwan for providing reagents. We thank Dr. Maureen Condic, Dr. Rodney Stewart, Dr. Marti Tristani for their review of this paper, and Yost lab members (in particular Angie Serrano) for discussions and input. This study was funded by a NHLBI Bench-to-Bassinet Consortium (http://www.benchtobassinet.com) grant to HJY (UM1HL098160). SAW was supported by National Institutes of Health under Ruth L. Kirschstein National Research Service Award 2T32HL007576-31 from the National Heart, Lung, and Blood Institute.

## Author contributions

S.A.W. conceived, designed, and conducted the experiments with input from H.J.Y. H.J.Y. and S.A.W. wrote the paper. BLD carried out statistical analysis and graphical presentation of data.

## Additional information

**Competing interests:** The authors declare no competing interests.

