## [Peer Review File · Nature Communications]

Reviewers' Comments:

Reviewer #1:

Remarks to the Author:

Major claims: The authors show that neural crest cells contribute to a unique cardiomyocyte population in the zebrafish ventricle. Elegant genetic experiments reveal that ablation of this NC-derived CM population leads to altered Notch signaling, hypertrabeculated hearts, altered cardiomyocyte size, diminished adult heart capacity and poor physiological response to cardiac stress. The authors claim that they have identified a novel developmental mechanism and genetic pathway that predisposes adults to hypertrophic cardiomyopathy and provides the first zebrafish model of adult-onset heart failure. Overall, the data are of very good quality and the transgenic lines generated to label and ablate at will specific cardiac populations are very useful for this, and future studies.

The main caveat with this MS is that some conclusions are not well substantiated and some claims should be re-elaborated.

Major comments:

-Page 5. The authors say that "Analysis of confocal slices of the ventricles from NC-Cm ablated hearts at 5dpf demonstrated an unusual disarray of trabeculation compared to control hearts (Fig.2E)". The images shown in Figure 2E are at very low magnification, would it be possible to show higher power images of the disarray in trabeculation? Would it help showing an H&E staining?

-Page 6. The authors use a Notch-GFP reporter to combine it with the Sox10 one and determine the spatial relationship between Notch-active and Sox10-expressing cells, as Notch-active cells do not trabeculate, and the Sox10-positive cell trabeculate and express jag2B. The authors ablate the Sox10 NC-CM lineage, isolate the hearts at 4dpf and make qPCR for Jag2B and Erbb2: Jag2B is downregulated, Erbb2 is unaffected. The authors say that Erbb2 is unaffected "perhaps because the expected increase in erbb2 expression, upon NC-Cm and Notch signaling disruption, is counteracted by the loss of erbb2 expression from the NC-Cms themselves in NC-Cm ablated hearts". Can the authors elaborate this sentence? Can the authors find a way to support this claim? In situ, a reporter for Erbb2? Can the authors show that Jag2B is expressed in NC-CM? Is there another way to show Notch activity in CM? Does the reporter line (TP1-GFP) reflects the real dynamics of Notch activity? Given that Notch activity in the embryonic mouse heart has not been reported in the myocardium, but in the endocardium, how sure are the authors about this CM-specific Notch activity in the embryonic ZF heart? Does Notch reporter driven GFP expression go down after Notch inhibition with DAPT or RO? Can the authors use a Notch target in the myocardium (hey? hesr?) as an indicator of Notch activity in non-trabeculating CM?

-The authors also say that (page 6-7): "Together, these data support and extend the current Notch/Neuregulin model of cardiac trabeculation, and lead us to propose that the NC-Cms are critical for the correct patterning of ventricular trabeculation by providing a unique cellular source of Jag2B (Figure 3C)". This statement would be better supported providing at least some of the data requested above.

-In page 7, the authors say that NC-Cm-ablated hearts show a "massive hypertrabeculation" (Figure 4F) and a two-fold decrease in the luminal area of the ventricle (Figure 4G). Also, CM in these NC-Cm-ablated hearts are significantly larger than in controls. All these are features of hypertrophic cardiomyopathy, rather than hypertrabeculation. Indeed, the images in Fig. 4D-F show that in controls hearts (D,E) there is the very dense normal trabecular network, while the NC-Cm-ablated hearts show a thicker network, compatible with straightforward hypertrophy phenotype, as the larger CM volume indicates. On this regard, reference 17 cited in page 10, is a review about the heterogeneity of left ventricular non-compaction cardiomyopathy (LVNC) but not on HCM. In LVNC, CM are not larger than in the normal heart, as the authors find here. In the last section of the paper, the authors describe the relevance of their model for understanding HCM and

heart failure, in syndromic and non-syndromic cases, that is a fair statement.

Reviewer #2:

Remarks to the Author:

The contribution of neural crest cells to cardiomyocyte formation is a curious and highly interesting process. While it has been documented well in previous studies, its significance for heart development and function has remained unclear. The novelty of this paper lies in the fact that the authors use combinatorial expression of transgenes to specifically ablate only the neural crest derived cardiomyocytes. This is an innovative method, and the reported consequences (trabeculation defects, hypertrophy and reduced cardiac function) are very interesting and provide important novel insight into the biology of the neural crest's contribution to heart development. Thus, I recommend publication of the paper, if the authors can address the following issues, which are mainly to strengthen the conclusions drawn from their data:

Major:

- 1) The authors should further verify the validity of their transgenic system for specific ablation of the NC-CMs. Although the *myl7* driven Cre responder line is expected to be only expressed in CMs, the authors should verify that ALL the traced cells they detect are indeed CMs by co-staining with a CM marker.
- 2) Figure 1: the authors count the number of cells traced by their system in the heart, and determine the relative volume occupied by these cells in the heart (around 15%). It's unclear how they then conclude that NC-derived cells constitute about 10% of the total CMs in the heart? They should count total CM number (GFP+) themselves and not rely on published data, in particular since there's a discrepancy with their volume assessment. Assessing the % of CMs labelled by their system is also important for comparison with previous lineage tracing data as published by Cavanaugh et al, *Developmental Biology* 404 (2015) 103–112.
- 3) Ext. Fig 2: Only few pacemaker cells are expected to exist and the *Isl* antibody has been described to only label few CMs in the embryonic heart at 2 dpf. However, in the author's stainings it looks like almost all CMs were positive, raising questions about the validity of their *Isl* immunofluorescence stainings. Authors should clarify how they can reliably detect pacemakers cells.
- 4) It would also be interesting to know whether adult fish derived from the NC-Cm ablated embryos have pacemaker defects. The authors could perform *Isl* stainings in adults and maybe even heart rate measurements in adults to look for arrhythmia?
- 5) Line 90: how do the authors know that "no new tagRFP+ CMs were observed"? Since they don't show us numbers on how many cells are left after the ablation, they can certainly not claim that there are no new ones appearing. Quantification of the number of NC-derived CMs found in MTZ treated hearts at different stages should be done both to quantify the efficacy of the ablation and to substantiate the claim in line 90.
- 6) Figure 3A. Images of the individual channels should be shown to allow the readers to judge themselves whether there is or is not co-expression of the 2 reporters. In addition, co-staining with a CM marker should be used to support the idea that *sox10*+ cells are CMs adjacent to *Tp1* reporter+ CMs.
- 7) The idea that NC-CMs are the (main) source of *Jag2b* ligand regulating trabeculation should be confirmed using *in situ* hybridization. Fluorescent *in situ* or RNAScope should give the necessary cellular resolution, and would allow for co-staining with the transgenic marker.

8) Figure 4. Individual channels should be shown of magnifications of panels H-J and examples pointed out of double positive nuclei that the authors counted as CMs and of non-CMs.

9) Extended Figure 7B: How many hearts were the dissociated cells derived from? Given the big variations in CMs sizes that likely occur naturally and that could be introduced by the dissociation/culture, several hearts should be analyzed and a total sample size of 25 cells appears low.

Minor:

Figure 4: The % of RFP+ CMs is missing in panel C (text states that FACS was also performed on these).

Figure 4K, L. the Term "UCrit" should be defined.

Reviewer #3:

Remarks to the Author:

In support of previous studies, Abdul-Wajid and colleagues report that neural crest cells give rise to approximately 10% of cardiomyocytes (CMs) in a developing zebrafish heart. They generated transgenic lines to specifically deplete this neural crest-derived cardiomyocytes (NC-CMs). Surprisingly, this did not lead to major immediate morphological or functional changes, i.e. in heart size and heart rate. However, subtle changes in heart trabeculation during development were associated with hypertrabeculation in the adult, a pathological phenotype reminiscent of some cardiomyopathies.

The study is potentially interesting, but is far too preliminary for publication in its present form. Without any deeper understanding of the molecular and cellular roles of NC-CMs, the data on the phenotype obtained upon NC-CM ablation are somewhat descriptive.

Specifically, the following issues need to be addressed:

1) The authors generate a new transgenic Sox10-Cre zebrafish line, which they propose is "exclusively" expressed in the neural crest lineage. This has to be shown. There is some evidence, also in other model systems, that Sox10 is (re-)expressed in cells of non-neural crest origin. Since this is a central point of the study, absence of Sox10 expression in the heart has to be demonstrated and the neural crest origin of the NC-CMs has to be shown unambiguously.

2) The authors propose a model, in which NC-CMs control trabeculation by influencing neighboring trabeculating CMs. This does not exclude, however, that NC-CMs themselves contribute to trabeculae. Can NC-CMs be traced to trabeculae in the developing and adult heart (or can the authors convincingly show that they can NOT be traced to trabeculae)? The authors allude to this possibility in Figure 1, but the number of "trabeculating NC-CMs" must be quantified.

3) In Figure 2, the trabeculation defect should be better demonstrated. Histological sections might help to better visualize the defect. Moreover, how can this phenotype potentially contribute to the hypertrabeculation seen in the adult?

4) The data presented in Extended Figure 2 have to be quantified to substantiate the hypothesis that NC-CMs contribute to pacemaker cells. Is pacemaker function impaired upon NC-CM ablation?

5) Despite NC-CM ablation, the authors found no significant effect on heart size. Why? Do non-NC CM compensate for the loss, e.g. by increased proliferation?

6) The data and model presented in Figure 3 are too preliminary. Given the potential contribution of NC-CMs to trabeculae (see my point 2) above), can the authors exclude that the phenotype is due to loss of trabeculating NC-CMs rather than impaired paracrine signaling? Furthermore, regulation of Notch by NC-CMs has to be shown more convincingly. To demonstrate expression of Jagged on NC-CMs, the cells have to be isolated (e.g. by FACS); decrease of Notch reported activity upon NC-CMs has to be shown with cellular resolution. Would specific ablation of Notch signaling in NC-CMs mimic cell depletion or could a rescue experiment with relevant ligands be performed in in vivo or in co-culture system?

7) In Figure 4, the authors propose that the number of cells is not changed (although Figure 4F suggests that, in total, there are many more cells in the NC-CM-ablated heart). The authors quantify the number of CMs per trabeculae area (and don't find an increase), but isn't the number of CMs increased overall? How are "trabeculae areas" defined?

8) The data presented in Extended Figure 7 are not convincing. The pictures in B don't seem to correspond to the quantification presented in C. 3D imaging/reconstruction in vivo would make a more compelling case for the proposed cellular mechanism underlying the phenotype.

9) In addition, why are there so many non-CM cells, in particular in the CM-ablated heart (Ext. Fig. 7B)?

Minor points:

Page 7, line 159 – reference to a figure should be made to back up the statement about decreased luminal area. Alternatively, the exact quantification has to be given in the text.

Supp. Fig.4 C: at which stage were the measurements performed?

Reference 9 is not displayed correctly in the reference list.

Reviewer #4:

Remarks to the Author:

The study by Abdul-Wajidet al investigated the role of cells from the neural crest in heart development. By using genetic and imaging tools, they show that ablation of neural crest derived cardiomyocytes causes hypertrabeculation and impairment of cardiac function under stress test. Additionally, they propose that the specific role of these cells from neural crest is to be a source of Jagged2, a ligand of the Notch receptor involved in the regulation of trabeculation.

This is a novel, well designed and conducted study that lays the ground for further studies on genetic variants and/or environmentally-driven perturbations during development affecting these specific cells as a possible cause of cardiomyopathies.

There are some issues that need to be addressed before considering the manuscript for publication:

1) The authors should provide more details (and references) on the rationale behind investigating Jagged2, and not other Notch ligands such Jagged1 or Delta-like ligand 1, also expressed in the heart (de La Pompa 2012) as the cause of hypertrabeculation.

2) The authors don't provide a direct evidence, such as any kind of immunodetection, that NC-Cms are the cells expressing JAG2B, or rather just promoting its expression in neighboring cardiomyocytes.

3) Other studies, also cited by the authors, have reported an interplay between Notch and ErbB2 in regulating trabeculation. In this study, Erbb2 expression was not significantly affected in NC-Cm ablated hearts compared to control hearts.

Based on this finding, any statement on erbb2 is only speculative and should not be included in the Results.

4) The sentence on page 10 lines 220-221 should be rephrased in order to refer more accurately to the different heart conditions. Heart failure is the clinical manifestation of heart damage due to

different causes, such as ischemic insult, cardiomyopathy or valve disease (Ferrari EHJ 2014). Similarly, non compaction left ventricle (NCLV), which can be symptomatic or not, is a distinct condition from hypertrophic cardiomyopathy (Arbustini 2017).

5) Related to point 4) while it seems accurate that NCLV cardiomyopathies have still unknown etiology, they should be differentiated from hypertrophic and dilated cardiomyopathies for which mutations have been identified.

6) The Results and Discussion sections should be separated.

7) Check the MS for incomplete sentences or missing words.

Reviewers' comments:

Reviewer #1 (Remarks to the Author):

Major claims: The authors show that neural crest cells contribute to a unique cardiomyocyte population in the zebrafish ventricle. Elegant genetic experiments reveal that ablation of this NC-derived CM population leads to altered Notch signaling, hypertrabeculated hearts, altered cardiomyocyte size, diminished adult heart capacity and poor physiological response to cardiac stress. The authors claim that they have identified a novel developmental mechanism and genetic pathway that predisposes adults to hypertrophic cardiomyopathy and provides the first zebrafish model of adult-onset heart failure. Overall, the data are of very good quality and the transgenic lines generated to label and ablate at will specific cardiac populations are very useful for this, and future studies. The main caveat with this MS is that some conclusions are not well substantiated and some claims should be re-elaborated.

RESPONSE: We appreciate the reviewer's enthusiastic support of our paper, and the comments that have helped to further substantiate and clarify the conclusions.

Major comments:

-Page 5. The authors say that "Analysis of confocal slices of the ventricles from NC-Cm ablated hearts at 5dpf demonstrated an 109 unusual disarray of trabeculation compared to control hearts (Fig.2E)". The images shown in Figure 2E are at very low magnification, would it be possible to show higher power images of the disarray in trabeculation? Would it help showing an H&E staining?

RESPONSE: We thank the reviewer for the positive and thorough review of the manuscript. We now present the fig2E microscopy images with a clearer color scheme and better resolution to address the reviewers' concern over its visual clarity.

-Page 6. The authors use a Notch-GFP reporter to combine it with the Sox10 one and determine the spatial relationship between Notch-active and Sox10-expressing cells, as Notch-active cells do not trabeculate, and the Sox10-positive cell trabeculate and express jag2B. The authors ablate the Sox10 NC-CM lineage, isolate the hearts at 4dpf and make qPCR for Jag2B and Erbb2: Jag2B is downregulated, Erbb2 is unaffected. The authors say that Erbb2 is unaffected "perhaps because the expected increase in erbb2 expression, upon NC-Cm and Notch signaling disruption, is counteracted by the loss of erbb2 expression from

the NC-Cms themselves in NC-Cm ablated hearts". Can the authors elaborate this sentence? Can the authors find a way to support this claim? In situ, a reporter for *ErbB2*? Can the authors show that *Jag2B* is expressed in NC-CM? Is there another way to show Notch activity in CM? Does the reporter line (TP1-GFP) reflects the real dynamics of Notch activity? Given that Notch activity in the embryonic mouse heart has not been reported in the myocardium, but in the endocardium, how sure are the authors about this CM-specific Notch activity in the embryonic ZF heart? Does Notch reporter driven GFP expression go down after Notch inhibition with DAPT or RO? Can the authors use a Notch target in the myocardium (hey? *hesr*?) as an indicator of Notch activity in non-trabeculating CM?

-The authors also say that (page 6-7):"Together, these data support and extend the current Notch/Neuregulin model of cardiac trabeculation, and lead us to propose that the NC-Cms are critical for the correct patterning of ventricular trabeculation by providing a unique cellular source of *Jag2B* (Figure 3C)". This statement would be better supported providing at least some of the data requested above.

RESPONSE: We have removed the ErBB2 qPCR result because of the confusion it causes in our model. We present two additional pieces of evidence for our model and the jag2b relation to NC-Cms.

1) We performed a cellular-resolution fluorescent in-situ hybridization for Jag2b in 3dpf hearts of our NC-Cm labelling line, in response to the reviewer's suggestion. We found a significant co-localization of jag2b expression within the NC-Cms (see new figure 3C and 3D as well as extended figure 6),

2) We genetically tested the role Jag2b. Strikingly, homozygous and heterozygous adult mutants have a severe adult onset cardiomyopathy, a phenotype similar to the NC-Cm ablated adults (new Extended fig.10).

The reviewer's questions about myocardial specific Notch activity, the Notch reporter transgene we used shows response in myocardial cells (Fig 3). The reviewer's question of whether DAPT treatment disrupts the Tp1 driven GFP has been addressed by Neil Chi's group in Han et al. Nature 2016. This paper and others in mammals have also demonstrated Notch effectors in the myocardium. We clarify this in the text and include additional references (de la Pompa and Epstein, 2012, and Lorentz et al. 2004) for the readers.

-In page 7, the authors say that NC-Cm-ablated hearts show a "massive hypertrabeculation" (Figure 4F) and a two-fold decrease in the luminal area of the ventricle (Figure 4G). Also, CM in these NC-Cm-ablated hearts are significantly larger than in controls. All these are features of hypertrophic

cardiomyopathy, rather than hypertrabeculation. Indeed, the images in Fig. 4D-F show that in controls hearts (D,E) there is the very dense normal trabecular network, while the NC-Cm-ablated hearts show a thicker network, compatible with straightforward hypertrophy phenotype, as the larger CM volume indicates. On this regard, reference 17 cited in page 10, is a review about the heterogeneity of left ventricular non-compaction cardiomyopathy (LVNC) but not on HCM. In LVNC, CM are not larger than in the normal heart, as the authors find here. In the last section of the paper, the authors describe the relevance of their model for understanding HCM and heart failure, in syndromic and non-syndromic cases, that is a fair statement.

RESPONSE: Done. We agree with the reviewer that the best description is hypertrophic cardiomyopathy (HCM), are grateful for these comments, and now emphasize this throughout the paper. We have had extensive discussions with clinical cardiologists about nomenclature. Our paper now includes additional approaches to quantify cardiomyocyte cell number and cell size in several approaches, driving home the conclusion of HCM. We have removed the LVNC reference and reworded our description of the adult phenotype accordingly. We also only refer to the phenotype now as hypertrophy in the adults and only reference hypertrophic cardiomyopathy papers/reviews.

--

Reviewer #2 (Remarks to the Author):

The contribution of neural crest cells to cardiomyocyte formation is a curious and highly interesting process. While it has been documented well in previous studies, its significance for heart development and function has remained unclear. The novelty of this paper lies in the fact that the authors use combinatorial expression of transgenes to specifically ablate only the neural crest derived cardiomyocytes. This is an innovative method, and the reported consequences (trabeculation defects, hypertrophy and reduced cardiac function) are very interesting and provide important novel insight into the biology of the neural crest's contribution to heart development. Thus, I recommend publication of the paper, if the authors can address the following issues, which are mainly to strengthen the conclusions drawn from their data:

RESPONSE: We appreciate the reviewer's enthusiastic support of our paper and have addressed the helpful points to strengthen the paper.

Major:

1) The authors should further verify the validity of their transgenic system for specific ablation of the NC-Cms. Although the myl7 driven Cre responder line is expected to be only expressed in CMs, the authors should verify that ALL the

traced cells they detect are indeed CMs by co-staining with a CM marker.

RESPONSE: Done. We have included (extended figure 3) high resolution microscopy showing an example of all ventricle NC-Cms and Cms co-labeling with MF20 (a zebrafish myosin antibody that shows distinct sarcomeric patterning in muscle tissue and cardiomyocytes). We add this information to the text description of the line as well.

2) Figure 1: the authors count the number of cells traced by their system in the heart, and determine the relative volume occupied by these cells in the heart (around 15%). It's unclear how they then conclude that NC-derived cells constitute about 10% of the total CMs in the heart? They should count total CM number (GFP+) themselves and not rely on published data, in particular since there's a discrepancy with their volume assessment. Assessing the % of CMs labelled by their system is also important for comparison with previous lineage tracing data as published by Cavanaugh et al, Developmental Biology 404 (2015) 103–112.

RESPONSE: Done. We have removed the volumetric measurements and, as suggested, added cell quantification. We used two methods, counting cells in confocal 3D stacks and FACS analysis of NC-Cm labelled hearts and GFP-labeled Cms at various time points in development and adults. Our double-transgenic system allows quantification of NC-Cms and gave percentages that were consistent with numbers obtained by using transgenes that labeled Neural Crest in multiple cell types (Cavanaugh et al. 2015). Ours is the first report of a persistent population of NC-Cms in adult hearts in any species.

3) Ext. Fig 2: Only few pacemaker cells are expected to exist and the Isl antibody has been described to only label few CMs in the embryonic heart at 2 dpf. However, in the author's stainings it looks like almost all CMs were positive, raising questions about the validity of their Isl immunofluorescence stainings. Authors should clarify how they can reliably detect pacemakers cells.

RESPONSE: Since the pacemaker was a minor point (and we didn't see any changes in heart rates), we removed text alluding to the pacemaker relation to NC-Cms. More importantly, we clarify that the Isl1/2 antibody we used marks the secondary heart field as seen in Extended Fig.4 and some of this secondary heart field marker overlaps with NC-Cm labelling.

4) It would also be interesting to know whether adult fish derived from the NC-Cm ablated embryos have pacemaker defects. The authors could perform Isl stainings in adults and maybe even heart rate measurements in adults to look for arrhythmia?

RESPONSE: We have removed the suggestion of pacemaker involvement. We had attempted to measure heart rates in adult fish of this line through echocardiography. We found there was a large variation in data acquisition even in wild-type adults. The dosing of anesthetic to conduct the measurements as well as the size of the adult fish are what obscured consistent data acquisition. However, our heart rate measurements at 6 and 14dpf did not find any significant differences (Extended Fig.6), suggesting that if there were any arrhythmia or heart rate differences in adulthood, they might be acquired or secondary to the development of the HCM.

5) Line 90: how do the authors know that "no new tagRFP+ CMs were observed"? Since they don't show us numbers on how many cells are left after the ablation, they can certainly not claim that there are no new ones appearing. Quantification of the number of NC-derived CMs found in MTZ treated hearts at different stages should be done both to quantify the efficacy of the ablation and to substantiate the claim in line 90.

RESPONSE: Done. We thank the reviewer for pointing this out and for the opportunity to clarify. We now include the FACS analysis and quantification of tagRFP+ in NC-Cm ablated adult ventricles – there is less than 1% tRFP+ in these ablated adult ventricles leading us to the conclusion that our ablation is highly effective and that there is no significant new tRFP+ cells post-embryonic stages and no denovo sox10 expression in the heart myocardium. We also now include the quantification of the tRFP+ cells in three different embryonic stages to support the conclusion that NC-Cms percentages are steady state after 2dpf.

6) Figure 3A. Images of the individual channels should be shown to allow the readers to judge themselves whether there is or is not co-expression of the 2 reporters. In addition, co-staining with a CM marker should be used to support the idea that sox10+ cells are CMs adjacent to Tp1 reporter+ CMs.

RESPONSE: Done, thanks for the suggestion. We modified Fig.3A to now include separate channel images as well as the MF20 staining in the same ventricles to show overlap with the sox10+ cells.

7) The idea that NC-CMs are the (main) source of Jag2b ligand regulating trabeculation should be confirmed using in situ hybridization. Fluorescent in situ or RNAScope should give the necessary cellular resolution, and would allow for co-staining with the transgenic marker.

RESPONSE: Done. We conducted a cellular resolution fluorescent in-situ hybridization for jag2b and include the data in Fig. 3C-D and Extended Figure 8. We refine our description of the results and conclusions to suggest that NC-Cms are a significant source of jag2b expression but not exclusive. In addition, we provide genetic evidence that jag2b gives an adult-onset cardiomyopathy phenotype similar to the heart phenotype of embryonically ablated NC-Cms (Extended Fig.10).

8) Figure 4. Individual channels should be shown of magnifications of panels H-J and examples pointed out of double positive nuclei that the authors counted as CMs and of non-CMs.

RESPONSE: Done. We increased magnification of the images that are now in Extended Fig.11 and include arrows and arrowheads to demarcate examples of double positive Mef2+DAPI+, as suggested.

9) Extended Figure 7B: How many hearts were the dissociated cells derived from? Given the big variations in CMs sizes that likely occur naturally and that could be introduced by the dissociation/culture, several hearts should be analyzed and a total sample size of 25 cells appears low.

RESPONSE: The data shown were amalgamations of N=4 experiments and each experimental condition included n=3-5 hearts. We now include this information in the figure legend and result description. While several hearts and experiments were conducted, we used stringent criteria that required only using single isolated (not clumps) and well-attached cardiomyocytes for accurate cell size measurements. Thus the minimum of 25 cells used are only cells that meet those metrics.

Minor:

Figure 4: The % of RFP+ CMs is missing in panel C (text states that FACS was also performed on these).

RESPONSE: Thanks, now included.

Figure 4K, L. the Term "UCrit" should be defined.

RESPONSE: Thanks, we now define UCrit in the text and Figure legend , and include the formula used in calculation in the methods section.

--

Reviewer #3 (Remarks to the Author):

In support of previous studies, Abdul-Wajid and colleagues report that neural crest cells give rise to approximately 10% of cardiomyocytes (CMs) in a developing zebrafish heart. They generated transgenic lines to specifically deplete this neural crest-derived cardiomyocytes (NC-CMs). Surprisingly, this did not lead to major immediate morphological or functional changes, i.e. in heart size and heart rate. However, subtle changes in heart trabeculation during

development were associated with hypertrabeculation in the adult, a pathological phenotype reminiscent of some cardiomyopathies.

The study is potentially interesting, but is far too preliminary for publication in its present form. Without any deeper understanding of the molecular and cellular roles of NC-CMs, the data on the phenotype obtained upon NC-CM ablation are somewhat descriptive.

Specifically, the following issues need to be addressed:

1) The authors generate a new transgenic Sox10-Cre zebrafish line, which they propose is “exclusively” expressed in the neural crest lineage. This has to be shown. There is some evidence, also in other model systems, that Sox10 is (re-)expressed in cells of non-neural crest origin. Since this is a central point of the study, absence of Sox10 expression in the heart has to be demonstrated and the neural crest origin of the NC-CMs has to be shown unambiguously.

RESPONSE: To address the reviewers concern of whether our transgenic sox10-cre is exclusive to neural crest, we provide a developmental series of Cre expression images from this line in Extended Figure 2. In early neural crest migratory stages (15-22hpf) we demonstrate by in-situ hybridization that cre expression from our Sox10-cre transgenic line marks the stereotypical neural crest migratory pattern. Furthermore, we show that this cre staining is not found EARLIER than 22hpf surrounding the developing heart tube as marked by myl7 expression. By 26hpf Cre switched cells in a different reporter line are found in the heart tube, similar to the results obtained by crossing this driver to our Cm:KillSwitch line. These results demonstrate the absence of sox10 expression in the heart (and heart field) prior the developmental time when neural crest cells arrive at the developing arches and proximal to the heart field. After NC-Cms incorporate into the heart we demonstrate, with additional quantification data from FACS (see new Figure 1), that the population of sox10 derived cardiomyocytes remains at constant percentage of the cardiomyocyte population into adulthood. The strongest piece of evidence that there is no de-novo sox10 expression in the heart is the absence of newly switched RFP cells after ablation of previously switched cells at 2dpf (see quantification in Figure 4C,D after ablation). While the transgenic constructs and lines are new, the promoter fragment used to make this transgene has been extensively characterized and used by at least three groups to demarcate neural crest in zebrafish. We now include citations for this sox10 promoter.

2) The authors propose a model, in which NC-CMs control trabeculation by influencing neighboring trabeculating CMs. This does not exclude, however, that NC-CMs themselves contribute to trabeculae. Can NC-CMS be traced to trabeculae in the developing and adult heart (or can the authors convincingly show that they can NOT be traced to trabeculae)? The authors allude to this possibility in Figure 1, but the number of “trabeculating NC-CMs” must be quantified.

RESPONSE: The reviewer is correct, the model does not exclude the possibility of NC-Cms contributing, along with neighboring Cms, to trabecula. This is the case. We now include Extended Fig. 3 that shows NC-Cms stained with MF20 antibody and clearly within the trabeculae of a 5dpf ventricle (similar to Fig.1J). This figure also includes quantification of the percent of NC-Cms found in trabeculae vs. the rest of the ventricle and this information was also added in the results section. We also now include sections of adult ventricles showing tRFP+ stained NC-Cms in the adult trabeculae (Extended Fig. 9) and the absence of tRFP+ cells in adults hearts from embryonic NC-CM ablation.

3) In Figure 2, the trabeculation defect should be better demonstrated. Histological sections might help to better visualize the defect. Moreover, how can this phenotype potentially contribute to the hypertrabeculation seen in the adult?

RESPONSE: Done. We have improved the images in Fig.2 to better visualize the trabeculation patterning defect. The question of how the embryonic trabeculation phenotype contributes to the adult phenotype, built on the foundation of this paper, is a great question that we plan to further delineate in future studies.

4) The data presented in Extended Figure 2 have to be quantified to substantiate the hypothesis that NC-CMs contribute to pacemaker cells. Is pacemaker function impaired upon NC-CM ablation?

RESPONSE: Our observation about pacemaker labeling was a minor point, so we removed it. The Isl1/2 antibody we used serves predominantly as a marker of secondary heart field. Because we found no significant differences in heart rate at 6 and 14dpf in NC-Cm ablated hearts, we think pacemaker function is likely intact (Extended Fig.6).

5) Despite NC-CM ablation, the authors found no significant effect on heart size. Why? Do non-NC CM compensate for the loss, e.g. by increased proliferation?

RESPONSE: Great question. We now include extended Fig.13 which demonstrates that the cardiomyocyte number at 5dpf is not significantly different. This is in addition to our adult cell quantification, showing comparable numbers of Cms, indicative of adult HCM. The likely explanation, as suggested by the reviewer, is that non-NC-derived cardiomyocytes fill in after ablation, which is known to happen in a more general (and larger) ablation of cardiomyocytes in zebrafish development (citations now included). Although we don't have data on proliferation rates, our observation that cardiomyocyte numbers are the same between control and ablated conditions suggest there is a compensatory mechanism at play in the heart. We discuss in the context of zebrafish's heart regeneration capabilities in the discussion. The heart can regenerate the number of cardiomyocytes but fails to regenerate/replace the unique functions of the NC-Cms.

6) The data and model presented in Figure 3 are too preliminary. Given the potential contribution of NC-CMs to trabeculae (see point 2) above), can the authors exclude that the phenotype is due to loss of trabeculating NC-CMs rather than impaired paracrine signaling? Furthermore, regulation of Notch by NC-CMs has to be shown more convincingly. To demonstrate expression of Jagged on NC-CMs, the cells have to be isolated (e.g. by FACS); decrease of Notch reported activity upon NC-CMs has to be shown with cellular resolution. Would specific ablation of Notch signaling in NC-CMs mimic cell depletion or could a rescue experiment with relevant ligands be performed in in vivo or in co-culture system?

RESPONSE: We now include the cellular resolution jag2b in-situ in the labelled NC-Cm embryos and demonstrate its preferential expression in NC-Cms. This concurs with our observation that Notch response pathway is upregulated in neighboring Cms, not in NC-Cms. To address the role of Jag2b, we added new genetic data showing that jag2b mutants have a phenotype similar to NC-Cm ablated adult hypertrophy (Extended Fig.10) thus strengthening our model, as the reviewer suggested, by 'mimic(ing) cell depletion' of NC-Cm jag2b.

7) In Figure 4, the authors propose that the number of cells is not changed (although Figure 4F suggests that, in total, there are many more cells in the NC-CM-ablated heart). The authors quantify the number of CMs per trabeculae area (and don't find an increase), but isn't the number of CMs increased overall? How are "trabeculae areas" defined?

RESPONSE: We provide further text in the methods section to clarify how we define the area in the ventricle that we are measuring. See response to points 8 and 9 below, outlining our multiple approaches that quantified Cm numbers in adult hearts.

8) The data presented in Extended Figure 7 are not convincing. The pictures in B don't seem to correspond to the quantification presented in C. 3D imaging/reconstruction in vivo would make a more compelling case for the proposed cellular mechanism underlying the phenotype.

RESPONSE: To clarify, we replaced the images of the chamber grown adult Cms with higher resolution microscopy images of single Cm isolates in each condition (See new Fig.4I-J). We also include movies of 3D reconstruction of these cells in Extended Movie 3 and 4.

9) In addition, why are there so many non-CM cells, in particular in the CM-ablated heart (Ext. Fig. 7B)?

RESPONSE: We replaced the pictures of the CM with representative single CM isolated cells to better represent the morphology difference we find in the adult CMs (see new Fig.4 I-J). Of

course, there will be many non-CM cells in culture isolates of adult hearts, which is why we think it is important to use the CM-specific GFP transgene to make sure we are measuring and quantifying cardiomyocytes in these mixed cultures and in FACS. Again, we present three different lines of evidence, FACS quantitation at adult (extended figure 12), immunofluorescent section quantification (extended fig.11) and embryonic whole heart microscopy and quantitation (extended fig.13) to demonstrate there is no significant difference in the number of cardiomyocytes between NC-ablated and control hearts.

Minor points:

Page 7, line 159 – reference to a figure should be made to back up the statement about decreased luminal area. Alternatively, the exact quantification has to be given in the text.

RESPONSE: Thanks, we reference and now include the exact numbers for the quantitative data in graph Fig.4H to back up the statement: “This measurement is inversely correlated with the luminal area that is available for blood flow through the ventricle, which was decreased two-fold, from 30% in controls to 13% in NC-Cm ablated hearts.”

Supp. Fig.4 C: at which stage were the measurements performed?

RESPONSE: Thanks, we added this information to the text.

Reference 9 is not displayed correctly in the reference list.

RESPONSE: Thanks, we fixed the reference.

--

Reviewer #4 (Remarks to the Author):

The study by Abdul-Wajidet al investigated the role of cells from the neural crest in heart development. By using genetic and imaging tools, they show that ablation of neural crest derived cardiomyocytes causes hypertrabeculation and impairment of cardiac function under stress test. Additionally, they propose that the specific role of these cells from neural crest is to be a source of Jagged2, a ligand of the Notch receptor involved in the regulation of trabeculation. This is a novel, well designed and conducted study that lays the ground for further studies on genetic variants and/or environmentally-driven perturbations during development affecting these specific cells as a possible cause of cardiomyopathies.

RESPONSE: Thank you for your enthusiastic support of our paper.

There are some issues that need to be addressed before considering the manuscript for publication:

1) The authors should provide more details (and references) on the rationale behind investigating Jagged2, and not other Notch ligands such Jagged1 or Delta-like ligand 1, also expressed in the heart (de La Pompa 2012) as the cause of hypertrabeculation.

RESPONSE: Han et al. 2006 show that jag2b is expressed in cardiomyocytes during trabeculation, so we tested whether NC-Cms serve as a source for Jag2b. Moreover, our newly added genetic data shows jag2b mutant adult ventricles mimic the adult ventricles from NC-Cm ablation in embryogenesis, further substantiating this Notch ligand as a key effector of the trabeculation regulatory pathway.

2) The authors don't provide a direct evidence, such as any kind of immunodetection, that NC-Cms are the cells expressing JAG2B, or rather just promoting its expression in neighboring cardiomyocytes.

RESPONSE: Thanks, we now include a fluorescent ISH for jag2b and show its preferential expression in NC-Cms in the ventricle. See new Fig.3 and extended fig8.

3) Other studies, also cited by the authors, have reported an interplay between Notch and ErbB2 in regulating trabeculation. In this study, Erbb2 expression was not significantly affected in NC-Cm ablated hearts compared to control hearts. Based on this finding, any statement on erbb2 is only speculative and should not be included in the Results.

RESPONSE: We agree and have now removed the ErBB2 qPCR data and text. The new fluorescent ISH data and functional genetic data now substantiate a focus on jag2b.

4) The sentence on page 10 lines 220-221 should be rephrased in order to refer more accurately to the different heart conditions. Heart failure is the clinical manifestation of heart damage due to different causes, such as ischemic insult, cardiomyopathy or valve disease (Ferrari EHJ 2014). Similarly, non compaction left ventricle (NCLV), which can be symptomatic or not, is a distinct condition from hypertrophic cardiomyopathy (Arbustini 2017).

RESPONSE: Thanks. We have had extensive discussions with clinical cardiologists about nomenclature. We agree with this reviewer and reviewer 1 that the best description is hypertrophic cardiomyopathy (HCM). We note that we clarified our quantification of cardiomyocyte cell number and cell size using several approaches. In this case, HCM gives rise to functional heart failure as assessed by diminished ability in swim tunnel tests. We have removed

the LVNC reference and reworded our description of the adult phenotype accordingly. We also only refer to the phenotype now as hypertrophy in the adults and only reference hypertrophic cardiomyopathy papers/reviews.

5) Related to point 4) while it seems accurate that NCLV cardiomyopathies have still unknown etiology, they should be differentiated from hypertrophic and dilated cardiomyopathies for which mutations have been identified.

RESPONSE: We agree, as discussed above, and have altered the text accordingly.

6) The Results and Discussion sections should be separated.

RESPONSE: We had formatted according to Nature style. The last two paragraphs are essentially discussion, and we can label it as such if the style editor allows.

Check the MS for incomplete sentences or missing words.

RESPONSE: Thanks, we hope we have caught typographical errors.

Reviewers' Comments:

Reviewer #1:

Remarks to the Author:

The authors have improved their revised MS substantially, including better images of the defective trabeculae in Fig. 2E, ISH for jag2b in Nc-Cm line showing co-localization in Fig. 3C, and have appropriately reinterpreted the adult cardiac phenotypes as HCM rather than LVNC.

About the questions on Notch reporter activity and its myocardial specificity, contrary to what the authors say, the paper by N Chi's group has not answered if the Notch-driven GFP expression goes down after Notch inhibition with DAPT or RO, nor showed the expression of a Notch target in the myocardium (hey? hesr?) as an indicator of Notch activity in non-trabeculating CM. This is why the Tp1 reporter is somewhat controversial. Regarding Notch activity in mammals, the published papers show Notch activity in the endocardium during chamber development (Grego-Bessa 2007; Luxán et al. 2013; VanDusen 2014; D'Amato 2016). The paper by de la pompa and Epstein is a review and not a primary research article. Concerning the reports showing Notch activity in the mouse myocardium, with the exception of a small cardiomyocyte population in the OFT expressing the Notch target Hes1 (Rochais et al., 2009), all the articles about Notch activity in the myocardium use ectopic expression approaches (N11CD; Mlc2v-cre) or LOF using DNMMML; MLC2v-cre, that may cause cardiac phenotypes that are not Notch-specific. At the moment, the more difficult task is to match the data available in the mouse about the role of Notch signaling sequentially, activated in the endocardium by endocardial or myocardial ligands, in ventricular trabeculation, compaction and cardiomyopathy, and the role of Notch signaling during zebrafish trabeculation. The proposed Notch-mediated lateral inhibition mechanism described in zebrafish is not completely clear, as various laboratories (including ours) have been unable to reproduce the hypertrabeculation phenotype associated with Notch inhibition. In any case, even if these data maybe of interest for the researchers interested in chamber development, the most important finding of this report is the contribution of a relatively small Sox10-derived CM population to the formation of the ventricular wall, whose elimination leads to HCM and heart failure.

Reviewer #2:

Remarks to the Author:

The authors have in a satisfactory manner responded to the issues I raised, except for one, which I however consider important:

In response to my point 5 asking for quantification of the ablation efficacy, the authors responded:

"Done. We thank the reviewer for pointing this out and for the opportunity to clarify. We now include the FACS analysis and quantification of tagRFP+ in NC-Cm ablated adult ventricles – there is less than 1% tRFP+ in these ablated adult ventricles leading us to the conclusion that our ablation is highly effective and that there is no significant new tRFP+ cells post-embryonic stages and no denovo sox10 expression in the heart myocardium. We also now include the quantification of the tRFP+ cells in three different embryonic stages to support the conclusion that NC-Cms percentages are steady state after 2dpf."

Where are these data? Where are the FACS data from adults? Do the authors expect me to search the entire manuscript for new data? I still cannot find any numbers on the embryonic ablation efficacy. Figure 2 just shows images and the text still states:

"Importantly, no new tagRFP+ cardiomyocytes were observed three days after the ablation period, but an occasional extruding remnant of a tagRFP+ NC-Cm was detected (Fig.2B-D)."

So it seems that the authors have not responded to my request to provide numbers on how many cells remain after the ablation or what "occasional" means. The authors have also not altered their phrasing "no NEW tagRFP+ cardiomyocytes". It's awkward to conclude that no new ones form without having first shown and stated that the ablation kills all cells...

If the authors add the quantification of ablation efficacy (or clarify where the data are that they claimed to have added), I recommend publication.

Reviewer #3:

Remarks to the Author:

This is a substantially revised version of a previous paper by Abdul-Wajid and colleagues on the role of cardiac neural crest cells in zebrafish heart development. The authors have addressed many of my previous concerns. As mentioned before, I find the study interesting as such. However, some important issues still remain open and should be addressed in a satisfactory manner before publication.

Below, I refer to my specific points raised concerning the originally submitted manuscript. All other points have been dealt with appropriately.

1) The authors generate a new transgenic Sox10-Cre zebrafish line, which they propose is "exclusively" expressed in the neural crest lineage. This has to be shown.

To address this point, the authors have generated a new Extended Figure 2, in which they aim to show absence of Sox10-Cre expression in early heart tube cells expressing myl7. However, the quality of panels A, A', B, and B' is not sufficient, the magenta background staining is much too high and it is, therefore, impossible to appreciate absence of myl7 expression in Sox10-Cre expressing cells. Thus, these results do NOT convincingly "demonstrate the absence of sox10 expression in the heart (and heart field) prior the developmental time when neural crest cells arrive at the developing arches and proximal to the heart field", as claimed.

2) The authors propose a model, in which NC-CMs control trabeculation by influencing neighboring trabeculating CMs. This does not exclude, however, that NC-CMs themselves contribute to trabeculae. Can NC-CMs be traced to trabeculae in the developing and adult heart (or can the authors convincingly show that they can NOT be traced to trabeculae)? The authors allude to this possibility in Figure 1, but the number of "trabeculating NC-CMs" must be quantified. The authors have now quantified NC-CMs contributing to trabeculae (Extended Fig.3). Accordingly, they state in their rebuttal letter that "The reviewer is correct, the model does not exclude the possibility of NC-Cms contributing, along with neighboring Cms, to trabecula. This is the case." Surprisingly, however, in the paper, the authors don't come back to these findings and stick to their model that the trabeculation defects seen upon NC-CM ablation (Figs 2 and 4) are due to cell non-autonomous effects involving Notch signaling normally regulated by NC-Cm (Fig. 3). While the latter might be true as well, the observed trabeculation defects could mainly result from absence of trabeculae-forming NC-CMs. This possibility has to be integrated in the model and needs to be discussed.

6) The data and model presented in Figure 3 are too preliminary. (...) Furthermore, regulation of Notch by NC-CMs has to be shown more convincingly. To demonstrate expression of Jagged on NC-CMs, the cells have to be isolated (e.g. by FACS); decrease of Notch reported activity upon NC-CMs has to be shown with cellular resolution. Would specific ablation of Notch signaling in NC-CMs mimic cell depletion or could a rescue experiment with relevant ligands be performed in vivo or in co-culture system?

The authors now nicely show that jag2b (i.e. Notch signaling) mutants exhibit a phenotype similar to NC-CMs-ablated fish. However, the data regarding expression of the Notch reporter Tp1:d2GFP (Fig. 3A) and jag2b (Fig. 3C) are still not convincing: In 3A, confocal images have to be shown at

cellular resolution. There seem to be several green cells (i.e. cells displaying Notch activity) that are also positive for TagRFP, which would contradict the authors' model (Notch activity is found in cells adjacent to NC-CMs, but not in NC-CMs themselves). In 3C, the dots reflecting jag2b mRNA expression do not seem to be specific for NC-CM cells, as claimed. For instance, the lower arrow points to a tag-RFP-positive cell, which exhibits as many (or as few) relatively weak yellow dots as several of the green cells. What are the many tag-RFP positive cells outside the ventricle (i.e. left of the of the ventricle marked by a dashed line)? Is tag-RFP expression found in cells other than ventricular NC-CM cells? Given that Figure 3 is central for the authors' model, its quality has to be improved and the findings need to be substantiated.

Further point: In the new Extended Figure 9B, the signal to noise ratio has to be improved. It is unclear from the available picture which cells are truly tagRFP-positive. As compared to the MTZ-treated sample, most cells of the DMSO-treated heart appear to display labeling above background. High magnification insets might help.

Reviewer #4:

Remarks to the Author:

The Authors have satisfactorily addressed all the issues I had raised. Additionally, the Authors should check Methods for experiments not included in the revised version (qRT_PCR for ErBB2). If the formatting is according to Nature style, I would leave it as it is.

RESPONSE TO EDITOR: Thank you for the opportunity to respond to the Referees and to revise our manuscript. We have highlighted the changes in the manuscript and give detailed responses below in *italics*.

Reviewers' comments:

Reviewer #1 (Remarks to the Author):

The authors have improved their revised MS substantially, including better images of the defective trabeculae in Fig. 2E, ISH for *jag2b* in Nc-Cm line showing co-localization in Fig. 3C, and have appropriately reinterpreted the adult cardiac phenotypes as HCM rather than LVNC.

About the questions on Notch reporter activity and its myocardial specificity, contrary to what the authors say, the paper by N Chi's group has not answered if the Notch-driven GFP expression goes down after Notch inhibition with DAPT or RO, nor showed the expression of a Notch target in the myocardium (hey? *hesr*?) as an indicator of Notch activity in non-trabeculating CM. This is why the *Tp1* reporter is somewhat controversial. Regarding Notch activity in mammals, the published papers show Notch activity in the endocardium during chamber development (Grego-Bessa 2007; Luxán et al. 2013; VanDusen 2014; D'Amato 2016). The paper by de la pompa and Epstein is a review and not a primary research article. Concerning, the reports showing Notch activity in the mouse myocardium, with the exception of an small cardiomyocyte population in the OFT expressing the Notch target *Hes1* (Rochais et al., 2009), all the articles about Notch activity in the myocardium use ectopic expression approaches (*N1ICD*; *Mlc2v-cre*) or LOF using *DNMAML*; *MLC2v-cre*, that may cause cardiac phenotypes that are not Notch-specific. At the moment, the more difficult task is to match the data available in the mouse about the role of Notch signaling sequentially, activated in the endocardium by endocardial or myocardial ligands, in ventricular trabeculation, compaction and cardiomyopathy, and the role of Notch signaling during zebrafish trabeculation. The proposed Notch-mediated lateral inhibition mechanism described in zebrafish is not completely clear, as various laboratories (including ours) have been unable to reproduce the hypertrabeculation phenotype associated with Notch inhibition. In any case, even if these data maybe of interest for the researchers interested in chamber development, the most important finding of this report is the contribution of a relatively small *Sox10*-derived CM population to the formation of the ventricular wall, whose elimination leads to HCM and heart failure.

RESPONSE: We thank the referee for their helpful and supportive review of this revision. First, we are not arguing against a role for Notch signaling also in the endocardium, and we now include the additional citations as well as the review article. We are just pointing out and focusing on notch signaling in myocardium. In fact, our Notch reporter also shows notch response in endocardium. However, as the referee recognizes, our important focus is on the small population of NC-derived myocardial cells, and our observations that Notch response in myocardium occurs not in these cells, but predominantly in myocardial cells adjacent to these cells. We have clarified these points in the text.

Second, we understand from the referee that there is controversy in that the inhibitor treatments used by the N Chi lab have not been reproduced by other labs. We address this in two ways.

[Redacted]

[Redacted]

Second, in considering the previous citation (Han et al 2016 Nature) that have raised controversy, we note that Han et al Extended Figure 2 (with particular attention to Panels F and M-P), indicated that the DAPT inhibitor was able to block expression of the transgene which we used. They show two different Tp1 reporter lines (eGFP and d2EGFP) overlap with myl7 expressing cells in the 72hpf (3dpf) zebrafish heart (white arrows in panels E, G, I and J). When they treated these two different Tp1 reporter lines with DAPT from 60-72hpf, there is a loss of eGFP, but much more thorough loss of d2GFP (destabilized GFP) in the myocardial cells (Han et al Extended Fig 2 panels F, H and N). Outside the heart, other labs have recapitulated DAPT treated loss of reporter expression from these Tp1 lines in different tissues (see the original paper that created the Tp1 line for zebrafish, by Parsons et al. 2009 Mech. Of Development, Figure 4A vs. 4B.)

We agree with the reviewer's discussion that evidence in mammals for myocardial notch activity is sparse, but we suggest that it may be largely overlooked due to expression in relatively few cells compared to strong detection of endocardial notch activity. We speculate that the differences between mammalian and zebrafish notch activity in cardiomyocytes and its role in trabeculation may be due to developmental timing and corresponding ventricle trabeculation maturation times. However, there are three citations which we have now added (Kratsios et al. 2010 Circ. Research, Collesi et al. 2008 JCB, Collesi et al. 2018) that report endogenous NICD detection via antibody staining in ventricle myocytes in embryonic, neonatal mice and rat ventricle. In particular, see Kratsios et al. 2009 Figure 1A and B and Collesi et. 2008 Figure 1G). While we cite these papers in our manuscript, it might be time for a short review article discussing these controversies and comparing/contrasting mammals and zebrafish!

--

Reviewer #2 (Remarks to the Author):

The authors have in a satisfactory manner responded to the issues I raised, except for one, which I however consider important:

In response to my point 5 asking for quantification of the ablation efficacy, the authors responded:

"Done. We thank the reviewer for pointing this out and for the opportunity to clarify. We now include the FACS analysis and quantification of tagRFP+ in NC-Cm ablated adult ventricles – there is less than 1% tRFP+ in these ablated adult ventricles leading us to the conclusion that our ablation is highly effective and that there is no significant new tRFP+ cells post-embryonic stages and no denovo sox10 expression in the heart myocardium. We also now include the quantification of the tRFP+ cells in three different embryonic stages to support the conclusion that NC-Cms percentages are steady state after 2dpf."

Where are these data? Where are the FACS data from adults? Do the authors expect me to search the entire manuscript for new data? I still cannot find any numbers on the embryonic ablation efficacy. Figure 2 just shows images and the text still states:

"Importantly, no new tagRFP+ cardiomyocytes were observed three days after the ablation period, but an occasional extruding remnant of a tagRFP+ NC-Cm was detected (Fig.2B-D)."

So it seems that the authors have not responded to my request to provide numbers on how many cells remain after the ablation or what "occasional" means. The authors have also not altered their phrasing "no NEW tagRFP+ cardiomyocytes". It's awkward to conclude that no new ones form without having first shown and stated that the ablation kills all cells...

If the authors add the quantification of ablation efficacy (or clarify where the data are that they claimed to have added), I recommend publication.

RESPONSE: We had added the quantification in Figure 4D in which we graphically display FACS data of the percentage of RFP+ NC-Cms left in adult hearts after embryonic MTZ treatment. We apologize for not

emphatically making this important point. We should have referenced this panel in the text, which we now have done (and highlighted the text changes for the reviewer). To make this point easier to find, we also have now added the actual FACS analysis plots that were used for quantification in Figure 4D, as a new Supplemental Fig.9, and cited them in the text.

Again, to reiterate we think the most convincing evidence of ‘no new tagRFP+ cardiomyocytes’ is the adult quantitation data in Figure 4C and D. But we do agree with the reviewer about that statement being awkward and have clarified our text on lines 93-97 (see highlighted).

We also apologize for not clarifying the embryonic ablation efficiency numbers. We now include the cell numbers in a graph in Supplemental Fig. 5 of the caspase staining in ablated hearts and quantification of the increased active caspase 3 staining relative to the controls (quantified in Supplemental Fig.5D). In addition, we now include a graphical quantitation of the number of RFP+ cells left in the ventricle at 5dpf, post ablation, indicative of the ‘remnant tagRFP + NC-Cms’ (see new Figure 2E). We thank the reviewer for these important points that now strengthen our conclusions that once the embryonic NC-Cms are ablated, they are not replaced by other sox10-expressing cardiomyocytes.

We thank the referee for helping bring out these important points.

--

Reviewer #3 (Remarks to the Author):

This is a substantially revised version of a previous paper by Abdul-Wajid and colleagues on the role of cardiac neural crest cells in zebrafish heart development. The authors have addressed many of my previous concerns. As mentioned before, I find the study interesting as such. However, some important issues still remain open and should be addressed in a satisfactory manner before publication.

Below, I refer to my specific points raised concerning the originally submitted manuscript. All other points have been dealt with appropriately.

1) The authors generate a new transgenic Sox10-Cre zebrafish line, which they propose is “exclusively” expressed in the neural crest lineage. This has to be shown.

To address this point, the authors have generated a new Extended Figure 2, in which they aim to show absence of Sox10-Cre expression in early heart tube cells expressing myl7. However, the quality of panels A, A', B, and B' is not sufficient, the magenta background staining is much too high and it is, therefore, impossible to appreciate absence of myl7 expression in Sox10-Cre expressing cells. Thus, these results do NOT convincingly “demonstrate the absence of sox10 expression in the heart (and heart field) prior the developmental time when neural crest cells arrive at the developing arches and proximal to the heart field”, as claimed.

RESPONSE: We have now provided higher magnification views of 22hpf in-situs for cre and myl7 so viewers can clearly see the relationship of cre expressing cells to myl7 cells are in mutually exclusive groups (see Supplemental Figure 2D,D', D''). In addition, at a slightly later stage (stage 26hpf) a few sox10 derived cells (sox10:cre line crossed with the ubi:switch reporter line) are beginning to appear in the heart field (between one to three positive cells) indicated at higher magnification in Supplemental Fig.2E'-F' by arrows.

2) The authors propose a model, in which NC-CMs control trabeculation by influencing neighboring

trabeculating CMs. This does not exclude, however, that NC-CMs themselves contribute to trabeculae. Can NC-CMs be traced to trabeculae in the developing and adult heart (or can the authors convincingly show that they can NOT be traced to trabeculae)? The authors allude to this possibility in Figure 1, but the number of “trabeculating NC-CMs” must be quantified.

The authors have now quantified NC-CMs contributing to trabeculae (Extended Fig.3). Accordingly, they state in their rebuttal letter that “The reviewer is correct, the model does not exclude the possibility of NC-CMs contributing, along with neighboring Cms, to trabecula. This is the case.” Surprisingly, however, in the paper, the authors don’t come back to these findings and stick to their model that the trabeculation defects seen upon NC-CM ablation (Figs 2 and 4) are due to cell non-autonomous effects involving Notch signaling normally regulated by NC-Cm (Fig. 3). While the latter might be true as well, the observed trabeculation defects could mainly result from absence of trabeculae-forming NC-CMs. This possibility has to be integrated in the model and needs to be discussed.

RESPONSE: The phenotype we see upon ablation of NC-Cms is not wholesale loss of trabeculae, but mis-patterning of trabeculae. Certainly, removal of some of the cells that contribute to trabeculae could contribute to this disarray. However, we note that total cardiomyocyte number rapidly returns to normal (lines 222-225, and Supplemental Fig.14). This is consistent with previous observations that arbitrary ablation of “large numbers of embryonic ventricular cardiomyocytes and reported no consequential effects on subsequent embryonic heart regeneration, function and trabeculation” (lines 220-222). Thus, we think the effect is greater than just the removal of a small percentage (12%) of cardiomyocytes. Our model proposes that cell non-autonomous regulation of neighboring cardiomyocytes contributes to the developmental decisions of placement of trabeculae. In response to this reviewer, we have clarified this in line 229-233: “While some of the trabeculae mis-patterning could be directly due to loss of NC-Cms that contribute to trabeculae, the enrichment of jag2b in NC-Cms and activation of Notch signaling in neighboring cells suggests a cell non-autonomous pathway that contributes to trabeculae patterning.”

6) The data and model presented in Figure 3 are too preliminary. (...) Furthermore, regulation of Notch by NC-CMs has to be shown more convincingly. To demonstrate expression of Jagged on NC-CMs, the cells have to be isolated (e.g. by FACS); decrease of Notch reported activity upon NC-CMs has to be shown with cellular resolution. Would specific ablation of Notch signaling in NC-CMs mimic cell depletion or could a rescue experiment with relevant ligands be performed in in vivo or in co-culture system?

The authors now nicely show that jag2b (i.e. Notch signaling) mutants exhibit a phenotype similar to NC-CMs-ablated fish. However, the data regarding expression of the Notch reporter Tp1:d2GFP (Fig. 3A) and jag2b (Fig. 3C) are still not convincing: In 3A, confocal images have to be shown at cellular resolution. There seem to be several green cells (i.e. cells displaying Notch activity) that are also positive for TagRFP, which would contradict the authors’ model (Notch activity is found in cells adjacent to NC-CMs, but not in NC-CMs themselves). In 3C, the dots reflecting jag2b mRNA expression do not seem to be specific for NC-CM cells, as claimed. For instance, the lower arrow points to a tag-RFP-positive cell, which exhibits as many (or as few) relatively weak yellow dots as several of the green cells. What are the many tag-RFP positive cells outside the ventricle (i.e. left of the of the ventricle marked by a dashed line)? Is tag-RFP expression found in cells other than ventricular NC-CM cells? Given that Figure 3 is central for the authors’ model, its quality has to be improved and the findings need to be substantiated.

RESPONSE: Regarding Figure 3A, we now include a 3D reconstruction of hearts (Supplemental Movie 2) for better visualization the relationship of the sox10+ cells to the gfp+ notch responsive cells. There are not ‘several’ cells that are both green and red positive, there is one rounded cell near the ventricle apex that partially overlaps with some green. As can be seen in the 3D reconstruction movie, this is an edge

effect and the apparent concurrence is instead due to the flattening of the z-stack of this confocal image: a green cell is extended behind and over a red cell (i.e. an example of a Notch responding green cell neighboring and in direct contact with a NC-Cm red cell). The 3D reconstruction (Supplemental Movie 2, please move the slider back and forth to see slow motion) clarifies this spatial relationship.

Regarding Figure 3C we have provided higher magnification images (now figure 3H-K) and a quantification of the in-situ results in the graph shown in Figure 3L, compiled from imaging multiple cells in multiple embryos, finding a significant enrichment in the amount of jag2b signal for NC-Cms versus Cms. We also include additional in situ images in Supplemental Fig.8. As mentioned, there is still some jag2B signal in the other Cms and we do not make the claim that Jag2b is exclusively expressed or 'specific' to the NC-Cms. We explicitly state that jag2b expression is enriched in the NC-Cms compared to other cardiomyocytes ($p=0.0011$), and now emphasize in response to the reviewer that there are lower levels of jag2b in other cardiomyocytes (line 140-142), and cite the quantification panel (Fig.3L). The enrichment of jag2b in NC-Cms concurs with our finding that there is a significant reduction of Jag2b when NC-Cms are ablated. Our model is that the NC-Cms are "providing a significant and stereotypically positioned source of Jag2B" and NOT that Jag2b is specific to the NC-Cms (line 143-144). The model (Figure 3M cartoon) has been clarified to indicate that in the presence of NC-Cms, they and neighboring non-NC-derived Cms form stereotypical trabeculae. In the absence of NC-Cms, trabeculae are formed by non-NC-derived Cms, but are disorganized.

Further point: In the new Extended Figure 9B, the signal to noise ratio has to be improved. It is unclear from the available picture which cells are truly tagRFP-positive. As compared to the MTZ-treated sample, most cells of the DMSO-treated heart appear to display labeling above background. High magnification insets might help.

RESPONSE: We include a higher magnification inset of Supplemental Fig.9b as requested (see now Supplemental Fig. 10C).

Reviewer #4 (Remarks to the Author):

The Authors have satisfactorily addressed all the issues I had raised. Additionally, the Authors should check Methods for experiments not included in the revised version (qRT_PCR for ErBB2). If the formatting is according to Nature style, I would leave it as it is.

RESPONSE: Thank you for your contributions that improved the presentation of our results. We have removed details from the Methods section that are no longer utilized in the manuscript.

Reviewers' Comments:

Reviewer #1:

Remarks to the Author:

The authors have improved substantially their MS and refined the Discussion about the role of Notch in ZF trabeculation and its potential implication in HCM. Concerning the references in the text, some of the papers cited to support the role of Notch in mammalian trabeculation are not correct. Thus, the primary research papers in mice are Grego-Bessa et al. Dev Cell 2007, Luxán et al Nat Med 2013 and D'Amato et al 2016, there is no need to cite a review. The papers by Collesi and Kratsios (refs. 15-17) do not support the statement on page 6 that "Notch signaling is an important regulator of trabeculation during heart development", as they refer to the effect of ectopic Notch expression in the fetal and adult heart.

In relation to the role of Notch in the mouse myocardium, the papers by Kratsios and Collesi report expression of active Notch in some cardiomyocytes, but their functional experiments show only Notch GOF data, and it is well-known in the Notch field that ectopic expression of the active receptor in any given tissue will lead to a phenotype related with deregulation of cell specification, proliferation, and/or survival.

Reviewer #2:

Remarks to the Author:

The authors have now clarified the last issue I had raised and I now recommend publication.

Reviewer #3:

Remarks to the Author:

In this re-revision, the authors have now addressed the remaining major issues.

However, with respect to my concerns regarding the transgenic lines used, I did not criticize the quality of Supplemental Figure 2D, but of A and B, revealing early expression of Sox10 and the (supposedly absent) expression of myl7. There is a clear staining in magenta (potentially reflecting myl7 expression) at early stages (compare 15/19 hpf with 22 hpf, when there is no overall magenta staining anymore). Since the lines used are non-inducible, early expression from the myl7 driver (even if aberrant) could lead to tracing of cells other than cardiomyocytes, including neural crest cells, despite mutually exclusive expression at later stages.

I assume that the authors consider the magenta staining to reflect unspecific background rather than broad myl7 expression and I trust that the authors have done appropriate experiments to prove this point. Therefore, I would at least mention in the corresponding figure legend that the magenta staining is considered to be unspecific.

(in the online Manuscript Items of NCOMMS-18-04060B, I cannot find the legends to the supplemental figures).

Response to Reviewers Comments

Reviewer #1 (Remarks to the Author):

The authors have improved substantially their MS and refined the Discussion about the role of Notch in ZF trabeculation and its potential implication in HCM. Concerning the references in the text, some of the papers cited to support the role of Notch in mammalian trabeculation are not correct. Thus, the primary research papers in mice are Grego-Bessa et al. Dev Cell 2007, Luxán et al Nat Med 2013 and D'Amato et al 2016, there is no need to cite a review. The papers by Collesi and Kratsios (refs. 15-17) do not support the statement on page 6 that "Notch signaling is an important regulator of trabeculation during heart development", as they refer to the effect of ectopic Notch expression in the fetal and adult heart.

In relation to the role of Notch in the mouse myocardium, the papers by Kratsios and Collesi report expression of active Notch in some cardiomyocytes, but their functional experiments show only Notch GOF data, and it is well-known in the Notch field that ectopic expression of the active receptor in any given tissue will lead to a phenotype related with deregulation of cell specification, proliferation, and/or survival.

-Removed Notch review citation

Reviewer #2 (Remarks to the Author):

The authors have now clarified the last issue I had raised and I now recommend publication.

Thanks.

Reviewer #3 (Remarks to the Author):

In this re-revision, the authors have now addressed the remaining major issues.

However, with respect to my concerns regarding the transgenic lines used, I did not criticize the quality of Supplemental Figure 2D, but of A and B, revealing early expression of Sox10 and the (supposedly absent) expression of myl7. There is a clear staining in magenta (potentially reflecting myl7 expression) at early stages (compare 15/19 hpf with 22 hpf, when there is no overall magenta staining anymore). Since the lines used are non-inducible, early expression from the myl7 driver (even if aberrant) could lead to tracing of cells other than cardiomyocytes, including neural crest cells, despite mutually exclusive expression at later stages.

I assume that the authors consider the magenta staining to reflect unspecific background rather than broad myl7 expression and I trust that the authors have done appropriate experiments to prove this point. Therefore, I would at least mention in the corresponding figure legend that the magenta staining is considered to be unspecific. (in the online Manuscript Items of NCOMMS-18-04060B, I cannot find the legends to the supplemental figures).

-Added recommended text to Supplementary Figure 2 legend.